# DYNAMEM: Consistent Long Video Generation via Hierarchical Memory and Motion Priors

**Jingyu Lin** [* 1]  **Xinyi Shang** [* 2 3]  **Peng Sun** [* 4 5]  **Cunjian Chen** [† 1]  **Zhiqiang Shen** [† 3]

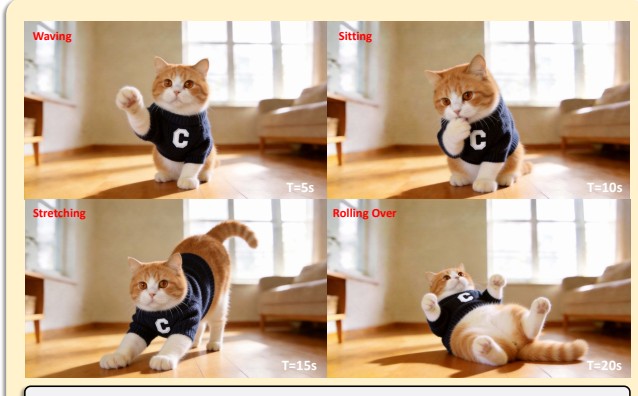 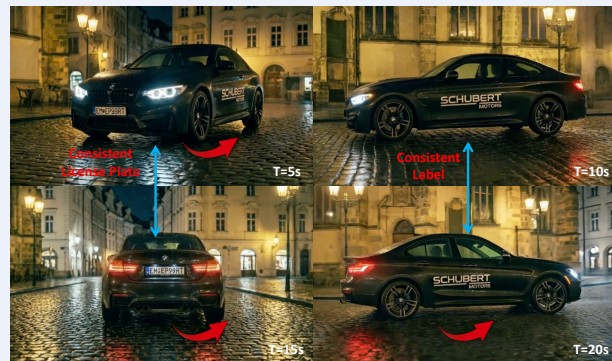

Figure 1. **Generated results of DYNAMEM via global prompt and local motion prompt.** Our DYNAMEM maintains superior object consistency while enabling smooth prompt-based control, achieving generation with little error accumulation. *Zoom in for better view.*

## Abstract

Recent text-to-video diffusion models can synthesize visually compelling clips from natural language prompts. However, practical applications increasingly demand long-form videos with evolving narratives and persistent identity. A common solution is autoregressive generation, where the video is produced clip by clip over long horizons, yet coherence often degrades as errors compound. In this work, we study long-video generation under an autoregressive setting, where videos are synthesized clip by clip over long horizons. Despite strong short-clip quality, existing approaches often suffer from semantic drift, motion decay, and appearance instability as the sequence grows. We present DYNAMEM, a unified framework that improves long-horizon coherence via three components: Semantic-Adaptive Hierarchical Memory for long-range semantic preservation, Motion-Prioritized Optimization for motion-coherent learning, and Reference-Anchored Perceptual Alignment for stabilizing appearance. Extensive experiments show that DYNAMEM produces more consistent semantics, stronger temporal dynamics, and more stable appearance on long videos compared to competitive baselines.

## 1. Introduction

Text-to-video diffusion models have advanced rapidly (Ho et al., 2020; Blattmann et al., 2023; Chen et al., 2024; Kong et al., 2024), yet real-world creation increasingly demands long-form videos with coherent narratives, persistent identities, and temporally consistent motion and appearance. In practice, such long videos are commonly generated autoregressively, where the model synthesizes each clip conditioned on a limited window of previous frames (Guo et al., 2025c; Lu et al., 2024; Team et al., 2025; Ma et al., 2026).

We refer to this setting as autoregressive long-video generation. Given a prompt, the goal is to synthesize a long video

---
*Equal contribution . †Corresponding authors. [1]Monash University [2]University College London [3]Mohamed bin Zayed University of Artificial Intelligence [4]Zhejiang University [5]Westlake University. Correspondence to: Cunjian Chen <cunjian.chen@monash.edu>, Zhiqiang Shen <Zhiqiang.Shen@mbzuai.ac.ae>.

*Proceedings of the 43rd International Conference on Machine Learning*, Seoul, South Korea. PMLR 306, 2026. Copyright 2026 by the author(s).

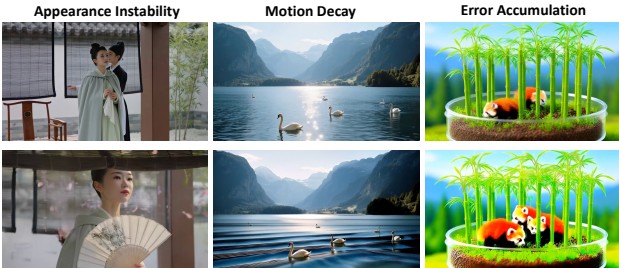

**Appearance Instability**    **Motion Decay**    **Error Accumulation**

*Figure 2.* **Failure cases of autoregressive long-video generation.** Autoregressive long video generation often faces three key challenges: (1) Appearance Instability; (2) Motion Decay, where motion gradually weakens and becomes static or less coherent; and (3) Error Accumulation. The three cases are taken from StoryMem (Zhang et al., 2025a), FramePack (Zhang et al., 2026), and CausVid (Yin et al., 2025), respectively.

by generating clips sequentially, where each step conditions on a constrained history due to computation and memory limits. In this long-horizon regime, coherence becomes the central bottleneck: local realism may remain strong, yet global consistency often collapses as the horizon grows.

As shown in Figure 2, despite impressive short-clip quality, existing autoregressive pipelines frequently fail in three intertwined ways over long horizons. First, **semantic drift** gradually alters early-established content such as subject identity, attributes, and scene layout. Second, **motion decay** accumulates across steps, causing dynamics to weaken and sometimes collapse into near-static imagery even when prompts describe ongoing actions. Third, **appearance instability** introduces color/illumination fluctuations that compound into perceptible flicker. These failures reinforce one another: once semantics begin to drift, both motion and appearance tend to degrade faster, and the errors compound across generation steps.

Achieving coherent long-video generation is challenging: attending to the full history is computationally infeasible while recent-frame conditioning alone cannot preserve distant semantics; standard training objectives are dominated by static regions, biasing learning toward copying context frames rather than modeling temporal change; and appearance drift from subtle distribution shifts is hard to correct without artifacts.

In this work, we propose **DYNAMEM**, a unified framework designed to improve long-horizon coherence by *jointly* addressing semantics, dynamics, and appearance (Figure 3). Our key insight is that long-video coherence cannot be reliably fixed by a single mechanism: semantic preservation, motion-coherent learning, and appearance stabilization must be improved in a coordinated yet lightweight manner that remains compatible with existing diffusion backbones.

First, we introduce **Semantic-Adaptive Hierarchical Memory (SAHM)** to preserve long-range semantics efficiently. SAHM maintains a sparse global reservoir of representative

frames and retrieves a compact working memory conditioned on the current prompt, allowing the generator to recall relevant content from distant timesteps without attending to the entire history.

Second, to mitigate motion decay, we propose **Motion-Prioritized Optimization (MPO)**, which strengthens temporal dynamics through (a) gradient-based data selection and (b) motion-weighted supervision that emphasizes dynamic regions. By prioritizing where and how the model learns motion, MPO counteracts the common bias toward static reconstruction and improves long-horizon temporal evolution.

Third, we stabilize appearance with **Reference-Anchored Perceptual Alignment (RAPA)**, a lightweight perceptual alignment that anchors each frame to a reference in CIELAB space with controlled blending. RAPA reduces accumulated color/illumination drift while avoiding excessive smoothing, resulting in more stable long-horizon appearance.

Extensive experiments on long-video generation validate that DYNAMEM produces more consistent semantics, stronger temporal dynamics, and more stable appearance over long horizons compared to competitive baselines. Notably, DYNAMEM remains effective even when prompts evolve over time, highlighting its robustness in realistic autoregressive generation settings.

Our main contributions are summarized as follows:

- We propose DYNAMEM, a comprehensive framework for autoregressive long-video generation, capable of mitigating semantic drift and motion decay issues, including scenarios with changing prompts.

- We introduce coordinated techniques spanning memory mechanisms, motion-prioritized optimization, and perceptual appearance anchoring, which synergistically enhance temporal coherence and appearance stability.

- Extensive experiments demonstrate that DYNAMEM achieves state-of-the-art performance, substantially improving long-horizon semantic consistency, motion strength, and appearance stability.

## 2. Related Work

**Video Generation.** Recent progress in video generation has been driven by large-scale video data and diffusion-based models (Ho et al., 2020). Early approaches (Chen et al., 2024; Guo et al., 2023; Blattmann et al., 2023) extend text-to-image diffusion models by augmenting U-Net backbones with temporal modeling, leveraging strong image priors and subsequent video fine-tuning. Diffusion models have also advanced identity-preserving generation (Lin et al., 2024b;a; Zhou et al., 2025a), cross-modal synthesis (Wang et al., 2024b), and scene-consistent storytelling (Song et al.,

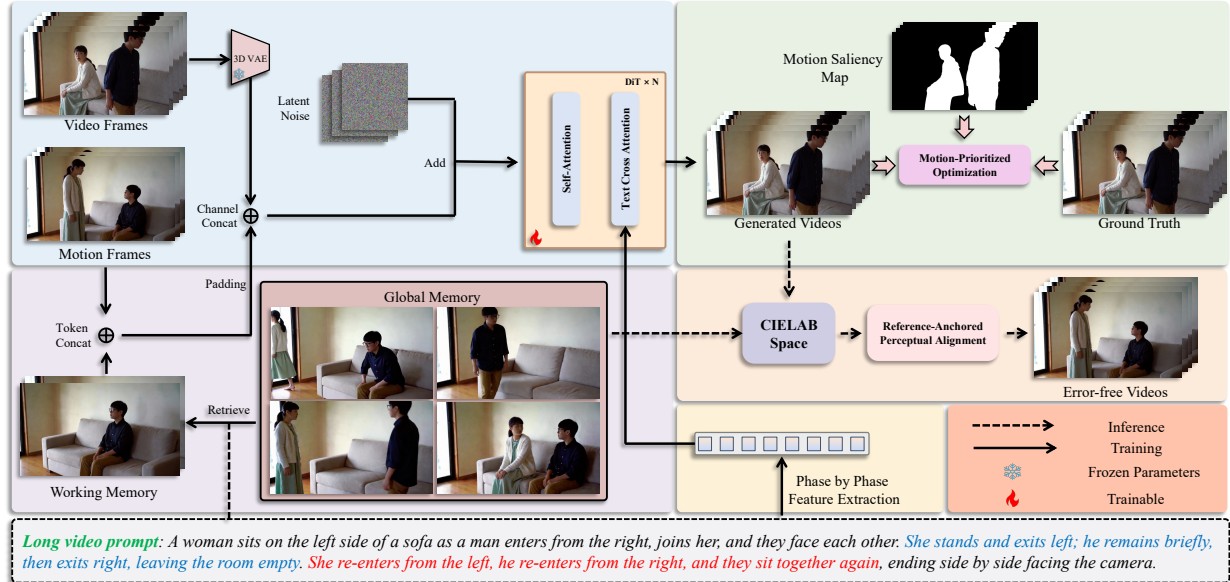

*Figure 3.* **Pipeline of DynaMem.** We enable consistent long-video generation by integrating hierarchical memory and motion-aware optimization into a DiT-based diffusion backbone. *(a) Semantic-Adaptive Hierarchical Memory* maintains long-term consistency by retrieving relevant historical frames from a global memory bank based on the current text prompt. *(b) Motion-Prioritized Optimization* counteracts motion decay by utilizing motion saliency maps to re-weight the training objective. *(c) Reference-Anchored Perceptual Alignment* stabilizes appearance over long horizons by aligning the perceptual color statistics of generated frames with a reference anchor in the CIELAB space.

2025b). More recent methods (Yang et al., 2025b; Kong et al., 2024; Wang et al., 2024a; Gao et al., 2025; Team et al., 2025; Li et al., 2024; Lin et al., 2025a; Zhou et al., 2026) adopt Diffusion Transformers (DiT), representing videos as sequences of spatiotemporal tokens. This formulation improves scalability and enables flexible generation across resolutions, aspect ratios, and video lengths. Complementary work on model compression (Ma et al., 2024a; 2023a;b; 2024b; Zheng et al., 2025) and efficient architecture design (Lin et al., 2024c) further improves the deployability of such large-scale models. Beyond visual synthesis, video diffusion models are increasingly used as world models for physical reasoning and simulation (He et al., 2025; Yu et al., 2025b; Song et al., 2025a), where long-horizon generation is essential. We adopt HunyuanVideo (Wu et al., 2025a) as our backbone for its strong motion modeling capabilities.

**Long Context Temporal Modeling.** Long Context Modeling is a core component of intelligent systems and has been extensively explored in large language models (Du et al., 2025; Wu et al., 2025b; Zhang et al., 2025c). More broadly, maintaining coherence over extended horizons is also essential in visual perception (Lin et al., 2023; 2022; Xu et al., 2025), knowledge transfer and model adaptation (Lin et al., 2026b;a; 2025b), and distributed learning (Guo et al., 2024b;a; 2025a;b). In video generation, however, explicit mechanisms for long-term memory remain relatively underdeveloped. Most existing approaches incorporate memory into video world models through inference-time opti-

mization (Hong et al., 2024), explicit 3D scene representations (Huang et al., 2025a; Li et al., 2025a), or retrieval-based cross-attention over past frames conditioned on camera motion or actions (Xiao et al., 2025; Yu et al., 2025a). These methods are primarily designed for controllable or interactive environments, where auxiliary signals such as camera poses or agent actions are available. As a result, their applicability to open-ended, text-driven video generation without external control remains limited.

## 3. Method

**Overview.** Figure 3 summarizes our approach to long-video generation, targeting semantic consistency, motion coherence, and appearance stability over long horizons. Section 3.1 reviews the diffusion formulation; Sections 3.2, 3.3, and 3.4 introduce our Semantic-Adaptive Hierarchical Memory, Motion-Prioritized Optimization, and Reference-Anchored Perceptual Alignment, respectively. Together, they enable coherent long-video generation without modifying the base diffusion backbone.

### 3.1. Preliminary

Video diffusion models aim to generate realistic video sequences by learning a gradual transformation from noise to data. Let $x_0 \in \mathbb{R}^{T \times H \times W \times C}$ denote a clean video with $T$ frames. Diffusion-based generative modeling constructs a continuous-time process $x_t$ where $t \in [0, 1]$, that interpo-

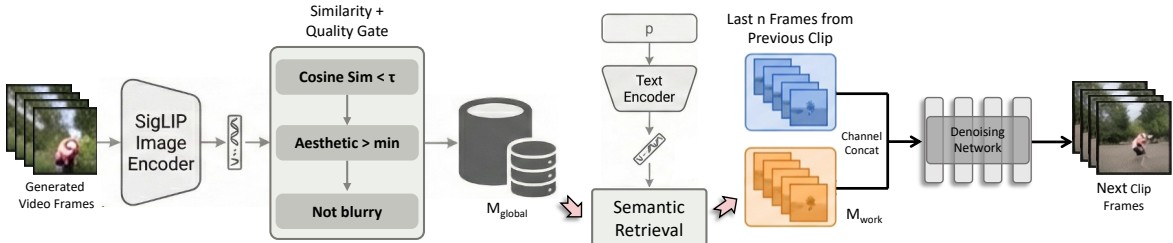

*Figure 4.* **Semantic-Adaptive Hierarchical Memory (SAHM).** We build a two-level memory for autoregressive long-video generation. Frames are encoded by SigLIP and sparsely stored in a global reservoir $\mathcal{M}_{global}$ via a similarity-and-quality gate. Given the prompt **p**, we retrieve top-$k$ relevant frames to form a working memory $\mathcal{M}_{work}$. $\mathcal{M}_{work}$ is concatenated with the last $n$ frames from the previous clip and injected into the denoiser to preserve long-range consistency. Analysis of extra computational cost can be found in Section A.3.

lates between the data distribution and an isotropic Gaussian prior. At training time, a noisy sample is obtained by combining the clean video with Gaussian noise $\varepsilon \sim \mathcal{N}(0, I)$,

$$x_t = \alpha(t)\,x_0 + \sigma(t)\,\varepsilon, \qquad (1)$$

where $\alpha(t)$ and $\sigma(t)$ are time-dependent coefficients controlling the signal-to-noise ratio. Specifically, a neural network (Peebles & Xie, 2023) $u_\theta$ is trained to approximate the temporal derivative of the noising trajectory,

$$u_\theta(x_t, t, c) \approx \frac{dx_t}{dt}, \qquad (2)$$

where $c$ denotes optional conditioning information. The learning objective is defined as a mean squared error between the predicted and target velocities,

$$\mathcal{L}_{\text{diff}} = \mathbb{E}_{x_0,\,\varepsilon,\,t}\big[\|u_\theta(x_t, t, c) - \dot{x}_t\|_2^2\big]. \qquad (3)$$

After training, novel videos are generated by initializing from Gaussian noise and numerically integrating the learned dynamics backward in time.

### 3.2. Semantic-Adaptive Hierarchical Memory

Autoregressive long-video generation often suffers from gradual loss of early visual information, such as character identity or scene layout. Naively extending the attention window to the entire history is computationally infeasible. To address this, we propose a Semantic-Adaptive Hierarchical Memory (SAHM) that maintains long-term consistency while keeping inference efficient. SAHM organizes memory into two levels, as shown in Figure 4: a global visual reservoir that stores the full generation history, and a compact working memory that is dynamically retrieved based on the current text prompt.

**Global Visual Reservoir.** We store generated frames in an external memory that does not consume GPU resources. To reduce redundancy, frames are archived only when they introduce new semantic content. Specifically, each frame is encoded with a SigLIP image encoder and compared to the most recent stored frame using cosine similarity. A

frame is added to the reservoir only if it meets a compound criterion: the similarity falls below a threshold $\tau$, while simultaneously satisfying a minimum aesthetic score and passing a blur detection check. This produces a sparse but representative global memory $\mathcal{M}_{global}$ that preserves the overall visual trajectory of the video.

**Prompt-Guided Working Memory.** At generation time, we construct a small working memory $\mathcal{M}_{work}$ by retrieving frames from $\mathcal{M}_{global}$ that are most relevant to the current Prompt **p**. We compute semantic similarity between the prompt embedding and all stored frame embeddings, and select the top-$k$ matches. This retrieval is updated whenever the prompt changes, allowing the model to recall relevant visual content from distant timesteps.

**Hybrid Context Injection.** To combine short-term continuity with long-term consistency, we inject two types of context into the denoising process: (1) the last $n$ frames from the previous clip for local transition stability, and (2) long-term semantic memory from $\mathcal{M}_{work}$ for global identity and scene cues. Both are concatenated with the input noise along the channel dimension.

### 3.3. Motion-Prioritized Optimization

Autoregressive video generation often suffers from motion decay, where dynamics degrade into static imagery over long horizons. We address this via a unified framework, Motion-Prioritized Optimization (MPO), which refines both data selection and the training objective.

**Gradient-Based Data Selection.** Standard filtering by motion magnitude often fails to distinguish meaningful object motion from camera shake. Instead, we employ influence functions to align the training distribution with high-quality dynamics. We define a small anchor set of clips by motion score and select training samples that maximize gradient similarity with these anchors. This ensures the model learns from data that contributes to coherent motion synthesis rather than static redundancy.

**Motion-Weighted Supervision.** Standard diffusion objectives are often dominated by static backgrounds, which can

**Algorithm 1** Semantic-Adaptive Hierarchical Memory

1: **Input:** text prompt sequence $\{\mathbf{p}_t\}$, clip length $\ell$, retrieval size $k$, similarity threshold $\tau$
2: Initialize global memory bank $\mathcal{M}_{global} \leftarrow \emptyset$
3: Initialize previous clip $\mathbf{V}_{prev} \leftarrow \emptyset$
4: **for** each generation step $t$ **do**
5:     Generate current clip $\mathbf{V}_t$ conditioned on $\mathbf{V}_{prev}$ and working memory
6:     Encode frames in $\mathbf{V}_t$ using SigLIP to obtain embeddings $\{\mathbf{e}_f\}$
7:     **for** each frame embedding $\mathbf{e}_f$ **do**
8:         **if** $\mathcal{M}_{global}$ is empty **or** $\text{sim}(\mathbf{e}_f, \mathbf{e}_{last}) < \tau$ **then**
9:             Add $(\mathbf{e}_f, f)$ to $\mathcal{M}_{global}$
10:         **end if**
11:     **end for**
12:     **if** prompt $\mathbf{p}_t$ changes **then**
13:         Retrieve top-$k$ frames from $\mathcal{M}_{global}$ by similarity to $\mathbf{p}_t$
14:         Construct working memory $\mathcal{M}_{work}$
15:     **end if**
16:     Form context by combining last $n$ frames of $\mathbf{V}_{prev}$ and $\mathcal{M}_{work}$
17:     Inject context into the denoising process for the next clip
18:     $\mathbf{V}_{prev} \leftarrow \mathbf{V}_t$
19: **end for**
20: **Output:** generated long video $\mathbf{V}$

cover over 90% of the video area. This imbalance causes conditional leakage, where the model over-relies on the context frame rather than learning temporal dynamics. To mitigate this, we introduce a spatially-adaptive loss. We compute a motion saliency map $\mathbf{M} \in [0,1]^{T \times H \times W}$ via optical flow and re-weight the objective to emphasize dynamic regions:

$$\mathcal{L}_{\text{total}} = \mathcal{L}_{\text{diff}} + \lambda \cdot \mathbb{E}_{t,\epsilon}\left[\|\mathbf{M} \odot (\epsilon_\theta(\mathbf{z}_t, \mathbf{c}, t) - \epsilon)\|_2^2\right] \quad (4)$$

By focusing optimization on high-motion areas, this mechanism counteracts motion decay and ensures consistent dynamics across sliding windows.

### 3.4. Reference-Anchored Perceptual Alignment

To stabilize long-horizon appearance, we align each generated frame's *perceptual color statistics* to a set of early, stable references drawn from the Global Memory. Instead of using a single reference frame, we take the **first three frames** stored in the global visual reservoir, denoted as $\mathbf{R}^{(i)}{}_{i=1}^{3}$, and perform perceptual moment alignment in CIELAB space (Reinhard et al., 2002). We choose CIELAB because it explicitly separates luminance (the $L$ channel) from chromatic components, which makes it convenient to correct color drift without over-altering brightness. Empirically, we find that

autoregressive long-horizon generation often suffers from a gradual increase in saturation and contrast, leading to an overly "greasy" appearance; operating in Lab space helps suppress this saturation buildup while preserving perceptual brightness and texture.

Given a generated chunk $\mathbf{X} \in \mathbb{R}^{3 \times T \times H \times W}$ with frames $\mathbf{x}_t$, we use $\phi(\cdot)$ and $\phi^{-1}(\cdot)$ for RGB↔Lab conversion and define $\mathbf{z}_t = \phi(\mathbf{x}_t)$. For each reference $\mathbf{R}^{(i)}$, let $\mathbf{z}_{\text{ref}}^{(i)} = \phi(\mathbf{R}^{(i)})$, and compute channel-wise spatial moments $\boldsymbol{\mu}_t, \boldsymbol{\sigma}_t \in \mathbb{R}^3$ from $\mathbf{z}_t$ and $\boldsymbol{\mu}_{\text{ref}}^{(i)}, \boldsymbol{\sigma}_{\text{ref}}^{(i)} \in \mathbb{R}^3$ from $\mathbf{z}_{\text{ref}}^{(i)}$.

We obtain a reference moment target by averaging the three early-memory statistics:

$$\boldsymbol{\mu}_{\text{ref}} = \frac{1}{3}\sum_{i=1}^{3} \boldsymbol{\mu}_{\text{ref}}^{(i)}, \qquad \boldsymbol{\sigma}_{\text{ref}} = \frac{1}{3}\sum_{i=1}^{3} \boldsymbol{\sigma}_{\text{ref}}^{(i)}. \quad (5)$$

Then, for each frame $t$ we apply closed-form affine moment alignment:

$$\hat{\mathbf{z}}_t = \frac{\boldsymbol{\sigma}_{\text{ref}}}{\boldsymbol{\sigma}_t + \varepsilon} \odot (\mathbf{z}_t - \boldsymbol{\mu}_t) + \boldsymbol{\mu}_{\text{ref}}, \quad (6)$$

where $\odot$ denotes channel-wise multiplication and $\varepsilon$ is a small constant. To avoid over-correction and preserve local content, we use residual blending in RGB space:

$$\mathbf{y}_t = (1 - \alpha)\mathbf{x}_t + \alpha\, \phi^{-1}(\hat{\mathbf{z}}_t), \qquad \alpha \in [0,1]. \quad (7)$$

In practice, we apply the transform on normalized RGB intensities with clamping, and handle near-zero $\boldsymbol{\sigma}_t$ with a safe fallback.

## 4. Experiments

### 4.1. Implementation Details

**Datasets.** To enhance training supervision for motion-intensive scenarios, we introduce **LM-100K**, a curated internal dataset comprising 100k high-quality long-take videos characterized by diverse physical motions. Each video is systematically partitioned into 5-second segments, which are subsequently annotated with fine-grained semantic captions using GPT-4o. These segments are organized into coherent sequences to provide robust temporal supervision for long-horizon and dynamic video generation. Further technical specifications of LM-100K are provided in Appendix A.1.

**Training setup.** We adopt a post-training strategy to fine-tune DYNAMEM on the LM-100K dataset, aiming to enhance its capability in generating long-take videos with coherent semantics and large-scale motions. The model is initialized with Hunyuanvideo-1.5 (Wu et al., 2025a), and trained end-to-end at a resolution of $854 \times 480$ and

*Table 1.* **Comparison of different methods on LV-Bench.** We report LV-Bench results on five VDE metrics and five complementary metrics from VBench (Huang et al., 2024). Our method achieves superior performance on the majority of these metrics.

| Method | LV-Bench (VDE) ↓ | | | | | VBench ↑ | | | | |
|---|---|---|---|---|---|---|---|---|---|---|
| | Subject | Background | Motion | Aesthetic | Clarity | Subject Consistency | Background Consistency | Motion Smoothness | Aesthetic Quality | Image Quality |
| MAGI-1 (Teng et al., 2025) | 0.3090 | 0.5000 | 0.0243 | 3.8286 | 2.7225 | 0.8992 | 0.9078 | 0.9947 | 0.6508 | 0.6662 |
| Self Forcing (Huang et al., 2025b) | 0.3716 | 1.6108 | 0.1549 | 3.4683 | 3.0798 | 0.8481 | 0.8203 | 0.9947 | 0.6283 | 0.6805 |
| PAVDM (Xie et al., 2025a) | 1.8292 | 0.9323 | 0.0461 | 2.8957 | 1.9503 | 0.8640 | 0.8924 | 0.9926 | 0.5267 | 0.6567 |
| FramePack (Zhang et al., 2026) | 4.3984 | 5.9421 | 0.0387 | 1.4751 | 4.2513 | 0.9001 | 0.8791 | 0.9949 | 0.6043 | 0.6972 |
| SkyReels-V2-DF (Chen et al., 2025) | 0.1085 | 0.3179 | 0.0195 | 1.2083 | 0.9365 | 0.9418 | 0.9579 | 0.9931 | 0.6035 | 0.6835 |
| BlockVid (Zhang et al., 2025b) | 0.0844 | 0.2945 | 0.0119 | 0.9618 | 0.7551 | 0.9597 | 0.9588 | 0.9956 | 0.6047 | 0.6852 |
| LongLive (Yang et al., 2025a) | 0.0827 | 0.3012 | 0.0105 | 0.9709 | **0.7291** | 0.9417 | 0.9542 | 0.9949 | 0.6053 | 0.6867 |
| DYNAMEM (Ours) | **0.0753** | **0.2732** | **0.0096** | **0.9524** | 0.7319 | **0.9677** | **0.9632** | **0.9956** | **0.6881** | **0.7082** |

$1280 \times 720$. All experiments are conducted on a distributed computing cluster equipped with high-performance NVIDIA H100 GPUs. We train the model on 32 GPUs using the AdamW optimizer with a stepwise decay schedule: the learning rate is initialized at $1 \times 10^{-4}$ and subsequently reduced to $3 \times 10^{-5}$, with weight decay set to $1 \times 10^{-4}$.

**Evaluation Metrics.** Drift-related penalties and evaluations are commonly used to quantify information dilution (Li et al., 2025b) and long-horizon degradation (Lu et al., 2024) in long video generation. Prior work has introduced identity-focused measures such as IP-FVR (Zhou et al., 2025b) and perceptual identity drift losses such as MoCA (Xie et al., 2025b). Inspired by LV-Bench (Zhang et al., 2025b), we adopt **Video Drift Error (VDE)** to characterize how video quality changes over time. Given a long video, we report five VDE-based metrics: (1) **VDE Clarity**, which captures sharpness degradation over time (higher indicates increasing blur); (2) **VDE Motion**, which reflects drift in motion smoothness (lower indicates stable dynamics with fewer jitters or freezes); (3) **VDE Aesthetic**, which measures temporal drift in overall visual appeal (lower indicates more consistent aesthetics); (4) **VDE Background**, which evaluates background/scene stability (lower indicates less flicker or setting drift); and (5) **VDE Subject**, which tracks identity consistency of the main subject across time (lower indicates less identity drift). Following prior works (Guo et al., 2025c; Cai et al., 2025; Zhang et al., 2025b), we additionally report five complementary metrics from VBench (Huang et al., 2024). The details are included in Appendix A.2.

*Table 2.* **Human evaluation.** We report vote rates (%) of DYNAMEM against each baseline. Evaluators choose {left, right, tie}. ID/Scene and Motion are attribute-specific votes.

| Compared method | Overall preference (%) | | | Attribute win (%) | |
|---|---|---|---|---|---|
| | Win | Tie | Lose | ID/Scene | Motion |
| BlockVid (Zhang et al., 2025b) | 63.9 | 15.0 | 21.1 | 69.5 | 67.8 |
| SkyReels-V2-DF (Chen et al., 2025) | 78.4 | 18.6 | 3.0 | 76.2 | 74.9 |
| LongLive (Yang et al., 2025a) | 55.1 | 20.2 | 24.7 | 63.8 | 61.7 |

### 4.2. Comparison with State-of-the-Art Methods

**Quantitative Comparison.** We compare DYNAMEM with several open-source long-video generation baselines on LV-Bench, including MAGI-1 (Teng et al., 2025), Self Forcing (Huang et al., 2025b), PAVDM (Xie et al., 2025a), FramePack (Zhang et al., 2026), SkyReels-V2-DF (Chen et al., 2025), LongLive (Yang et al., 2025a), and BlockVid (Zhang et al., 2025b). As shown in Table 1, DYNAMEM achieves the best performance on long-horizon degradation (VDE), indicating stronger robustness to error accumulation in autoregressive generation. The largest gains are on subject and background degradation, showing better identity and scene-layout preservation over time. DYNAMEM also improves motion degradation, consistent with our goal of reducing motion decay, while remaining competitive on clarity, suggesting stability improvements do not sacrifice sharpness. Beyond degradation, DYNAMEM delivers the highest overall quality on VBench, with consistent improvements across subject/background consistency, motion smoothness, and aesthetic quality, confirming the synergy of memory-based retrieval, motion-prioritized learning, and appearance stabilization.

**Human Evaluation.** To measure human preference, we conduct a side-by-side. We recruit 37 human evaluators who receive brief training and calibration examples to ensure a consistent understanding of the evaluation criteria. We evaluate a randomly selected subset of 20 prompts, with videos presented in randomized order and without method identifiers. For each pair, evaluators provide an overall preference, and additionally indicate which result exhibits better (i) identity/scene consistency and (ii) motion smoothness, with a "tie" option when differences are indistinguishable. As shown in Table 2, DYNAMEM is preferred in the majority of cases across all comparisons, with the largest margins on identity/scene consistency and motion smoothness.

**Qualitative Comparison.** Figure 5 compares long-horizon generations under the same prompt. Existing long-video baselines exhibit different forms of temporal degradation as

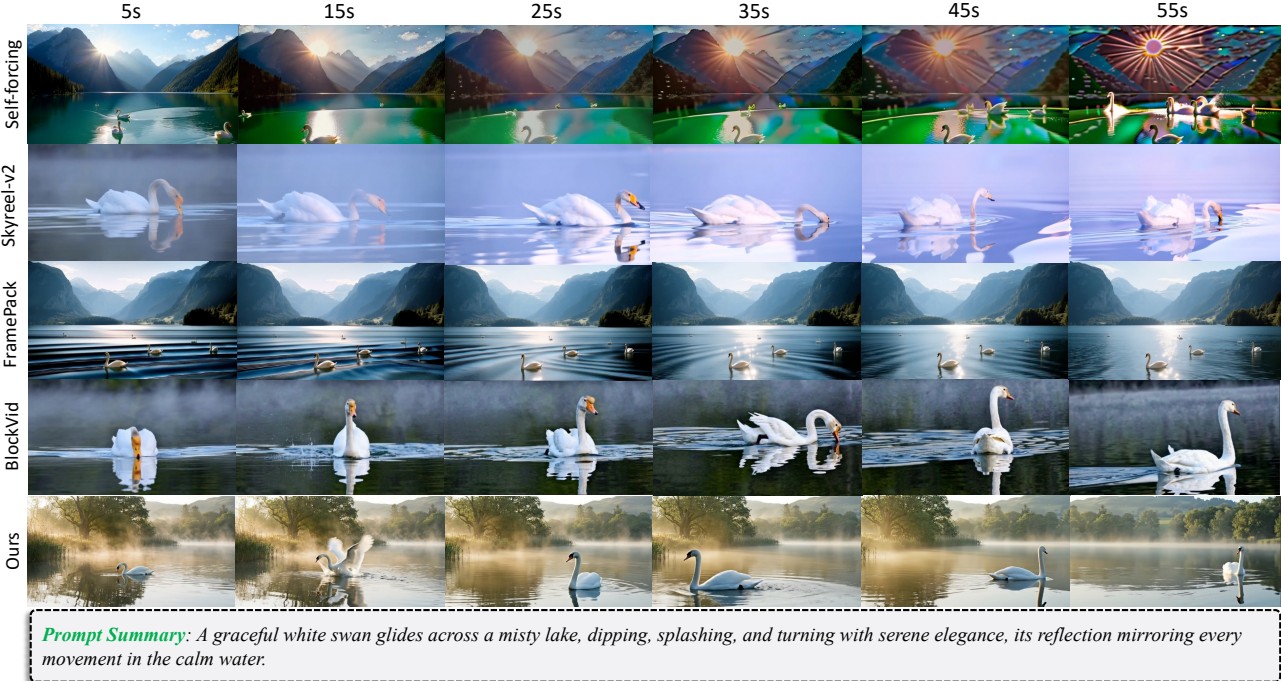

*Figure 5.* **Qualitative comparison.** Existing long-video baselines exhibit varying degrees of error accumulation over time, leading to motion decay and limited/fragmented dynamics. In contrast, our method effectively mitigates these failures, maintaining coherent appearance and temporally consistent motion while producing larger-amplitude and smoother actions. *Zoom in for a better view.*

the horizon extends. *Self-forcing* quickly accumulates errors and drifts into severe visual artifacts and unrealistic patterns, indicating unstable long-range propagation. *SkyReels-V2* shows evident motion decay: the swan's movement becomes increasingly weak and blurry, with partial deformation and loss of fine-grained dynamics. *FramePack* better preserves the overall scene but produces limited motion, where the swan largely remains in similar poses and the intended action evolution is under-expressed. *BlockVid* can generate noticeable motion early on, yet later frames reveal inconsistency and degradation (abrupt pose changes and weakened temporal coherence), suggesting error accumulation over time. In contrast, DYNAMEM maintains a stable scene and coherent swan identity throughout the sequence, demonstrating improved robustness against error accumulation and motion decay in long-horizon generation.

### 4.3. Ablation Study and Discussion

We conduct ablation studies, unless otherwise stated, on 8 GPUs and ensure fair comparison by using the same backbone, training schedule, and evaluation protocol.

**Ablation of Memory Modules.** We evaluate several memory variants to isolate the contribution of long-term retrieval and memory curation. As shown in Table 3, using only short-term context (*w* last frames) yields substantially worse degradation scores, indicating that local continuity alone

cannot prevent long-horizon drift in identity and scene layout. Replacing it with motion-driven references (*w* motion frames) improves both VDE and VBench, suggesting that emphasizing dynamic content helps mitigate motion decay and reduces static collapse. A stronger baseline is to retrieve from the full history without filtering (*w* all frames), which further improves consistency; however, it remains inferior to DYNAMEM and is less scalable due to uncurated growth of the reference set. In contrast, DYNAMEM achieves the best performance on all VDE metrics and the highest overall VBench quality. Notably, the largest gains appear in subject/background degradation and consistency, validating that prompt-guided semantic retrieval from a curated global reservoir is critical for preserving identity and scene layout over long horizons.

**Ablation of Motion-Prioritized Optimization.** As shown in Table 4, optimizing with $\mathcal{L}_{\text{diff}}$ only leads to clear motion decay: the model tends to preserve static backgrounds while dynamics gradually vanish, reflected by poor motion-related performance and noticeably degraded background/clarity. Adding motion-aware SFT already improves long-horizon dynamics and brings consistent gains on VBench. In contrast, DYNAMEM achieves the best results across both benchmarks with a much larger margin, especially on motion and temporal consistency, while simultaneously improving subject/background consistency and overall image quality. These results confirm that MPO is most effective

*Table 3.* **Ablation of Memory Modules.** We compare different video continuation settings. *w* last frame uses only the last frame from the previous clip; *w* motion frames additionally selects 8 frames as references; *w* all frames retrieves from all past frames.

| Variant | LV-Bench (VDE) ↓ | | | | | VBench ↑ | | | | |
|---|---|---|---|---|---|---|---|---|---|---|
| | Subject | Background | Motion | Aesthetic | Clarity | Subject Consistency | Background Consistency | Motion Smoothness | Aesthetic Quality | Image Quality |
| *w/* last frame | 1.2190 | 1.5629 | 0.0948 | 3.3281 | 1.9120 | 0.8616 | 0.8839 | 0.9926 | 0.5504 | 0.6542 |
| *w/* motion frames | 0.6547 | 0.9833 | 0.0619 | 2.1574 | 1.4092 | 0.9128 | 0.9156 | 0.9942 | 0.5987 | 0.6713 |
| *w/* all frames | 0.1845 | 0.4521 | 0.0142 | 1.1109 | 0.8856 | 0.9505 | 0.9412 | **0.9961** | 0.6534 | 0.6905 |
| DYNAMEM (Ours) | **0.0753** | **0.2732** | **0.0096** | **0.9524** | **0.7319** | **0.9677** | **0.9632** | 0.9956 | **0.6881** | **0.7082** |

*Table 4.* **Ablation of Motion-Prioritized Optimization.** We compare training variants for improving long-horizon dynamics. $w$ $\mathcal{L}_{\text{diff}}$ only further adds the standard diffusion objective. *add* motion sft applies motion-aware supervised fine-tuning with the selected data.

| Variant | LV-Bench (VDE) ↓ | | | | | VBench ↑ | | | | |
|---|---|---|---|---|---|---|---|---|---|---|
| | Subject | Background | Motion | Aesthetic | Clarity | Subject Consistency | Background Consistency | Motion Smoothness | Aesthetic Quality | Image Quality |
| *w/* $\mathcal{L}_{\text{diff}}$ only | 0.4125 | 0.7842 | 0.0537 | 1.2568 | 1.7432 | 0.8856 | 0.8923 | 0.9887 | 0.6514 | 0.6235 |
| *add* motion sft | 0.3429 | 0.5354 | 0.0243 | 1.0094 | 1.4689 | 0.9014 | 0.9165 | 0.9908 | 0.6798 | 0.6451 |
| DYNAMEM (Ours) | **0.0753** | **0.2732** | **0.0096** | **0.9524** | **0.7319** | **0.9677** | **0.9632** | **0.9956** | **0.6881** | **0.7082** |

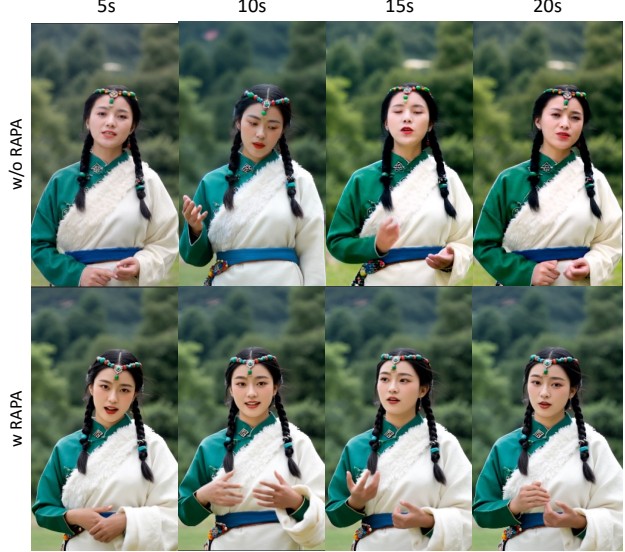

*Figure 6.* **Ablation of Reference-Anchored Perceptual Alignment.** RAPA stabilizes perceptual attributes across frames, preserving consistent identity in long-horizon generation.

when combining gradient-based data selection with motion-weighted supervision, yielding stable and coherent dynamics over long horizons rather than merely increasing motion magnitude.

**Ablation of Reference-Anchored Perceptual Alignment.** Figure 6 illustrates that RAPA stabilizes appearance in autoregressive long-horizon generation. Without RAPA (top row), small reconstruction errors accumulate over rollouts, leading to a noticeable drift in color statistics (increasing saturation/contrast) and gradual identity changes in facial details. With RAPA enabled (bottom row), the generation

*Table 5.* **IQA over time on 1-minute generations.** We compare models trained *w* vs. *w/o* RAPA module.

| Method | 10s | 20s | 30s | 40s | 50s | 60s |
|---|---|---|---|---|---|---|
| w/o RAPA | 1.98 | 1.95 | 1.88 | 1.81 | 1.74 | 1.67 |
| w/ RAPA | 1.99 | 1.98 | 1.98 | 1.97 | 1.98 | 1.97 |

remains anchored to the reference perceptual features, preserving consistent tones, textures, and facial identity across time. Overall, RAPA effectively reduces error accumulation and improves long-range identity consistency.

As a quantitative supplement, we compare 1-minute generations from models trained *with* and *without* RAPA using the image-quality metric IQA↑. We compute IQA over time by sampling frames at different timestamps. As shown in Table 5, IQA for the model *without* RAPA begins to drop after ∼10 seconds, whereas the RAPA-enabled model stays largely stable over the full minute, confirming improved long-horizon appearance consistency.

## 5. Conclusion

In this work, we introduced DYNAMEM, a unified framework that jointly improves semantic preservation, temporal dynamics, and visual stability without modifying the underlying diffusion backbone. By integrating semantic-adaptive memory retrieval, dynamics-prioritized optimization, and reference-anchored perceptual alignment, our approach enables more coherent and stable video generation over extended durations. Experimental results demonstrate that DYNAMEM consistently outperforms strong baselines in maintaining semantic consistency, sustaining motion, and stabilizing appearance in long videos.

## Acknowledgments

This research is supported by the Faculty Initiatives Research of Monash University (Contract No. 2901912), by the NVIDIA Academic Hardware Grant Program, and by the MBZUAI-WIS Joint Program for Artificial Intelligence Research.

## Impact Statement

While our primary goal is to push the boundaries of video generation research, we acknowledge the potential societal implications, including both positive impacts, such as enhancing accessibility in digital content creation and education, and risks, such as misuse for generating deceptive or harmful content. To mitigate this, we advocate for responsible use and the development of safeguards to ensure this technology serves constructive societal purposes.

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

Contributors., P. Pyscenedetect. https://www.scenedetect.com.

De Myttenaere, A., Golden, B., Le Grand, B., and Rossi, F. Mean absolute percentage error for regression models. *Neurocomputing*, 192:38–48, 2016.

Du, Y., Huang, W., Zheng, D., Wang, Z., Montella, S., Lapata, M., Wong, K.-F., and Pan, J. Z. Rethinking memory in ai: Taxonomy, operations, topics, and future directions. *arXiv e-prints*, pp. arXiv–2505, 2025.

Gao, X., Hu, L., Hu, S., Huang, M., Ji, C., Meng, D., Qi, J., Qiao, P., Shen, Z., Song, Y., et al. Wan-s2v: Audio-driven cinematic video generation. *arXiv preprint arXiv:2508.18621*, 2025.

Guo, S., Wang, H., and Geng, X. Dynamic heterogeneous federated learning with multi-level prototypes. *Pattern Recognition*, 153:110542, 2024a.

Guo, S., Wang, H., Lin, S., Kou, Z., and Geng, X. Addressing skewed heterogeneity via federated prototype rectification with personalization. *IEEE Transactions on Neural Networks and Learning Systems*, pp. 1–13, 2024b.

Guo, S., Lv, J., and Geng, X. Harmonizing generalization and personalization in ring-topology decentralized federated learning. *arXiv preprint arXiv:2504.19103*, 2025a.

Guo, S., Lv, J., Wang, Q., and Geng, X. Gene-fl: Gene-driven parameter-efficient dynamic federated learning. *arXiv preprint arXiv:2504.14628*, 2025b.

Guo, Y., Yang, C., Rao, A., Liang, Z., Wang, Y., Qiao, Y., Agrawala, M., Lin, D., and Dai, B. Animatediff: Animate your personalized text-to-image diffusion models without specific tuning. *arXiv preprint arXiv:2307.04725*, 2023.

Guo, Y., Yang, C., Yang, Z., Ma, Z., Lin, Z., Yang, Z., Lin, D., and Jiang, L. Long context tuning for video generation. *arXiv preprint arXiv:2503.10589*, 2025c.

Han, W., Lin, W., Zhou, Y., Liu, Q., Wang, S., Yao, C., and Chen, J. Show and polish: reference-guided identity preservation in face video restoration. *arXiv preprint arXiv:2507.10293*, 2025.

He, X., Peng, C., Liu, Z., Wang, B., Zhang, Y., Cui, Q., Kang, F., Jiang, B., An, M., Ren, Y., et al. Matrix-game 2.0: An open-source real-time and streaming interactive world model. *arXiv preprint arXiv:2508.13009*, 2025.

Ho, J., Jain, A., and Abbeel, P. Denoising diffusion probabilistic models. *Advances in neural information processing systems*, 33:6840–6851, 2020.

Hong, Y., Liu, B., Wu, M., Zhai, Y., Chang, K.-W., Li, L., Lin, K., Lin, C.-C., Wang, J., Yang, Z., et al. Slowfast-vgen: Slow-fast learning for action-driven long video generation. *arXiv preprint arXiv:2410.23277*, 2024.

Huang, J., Hu, X., Han, B., Shi, S., Tian, Z., He, T., and Jiang, L. Memory forcing: Spatio-temporal memory for consistent scene generation on minecraft. *arXiv preprint arXiv:2510.03198*, 2025a.

Huang, T., Zhou, C., Yao, X., Zhang, R.-X., Wu, C., Yu, B., and Sun, L. Quality-aware neural adaptive video streaming with lifelong imitation learning. *IEEE Journal on Selected Areas in Communications*, 38(10):2324–2342, 2020.

Huang, X., Li, Z., He, G., Zhou, M., and Shechtman, E. Self forcing: Bridging the train-test gap in autoregressive video diffusion. *arXiv preprint arXiv:2506.08009*, 2025b.

Huang, Z., He, Y., Yu, J., Zhang, F., Si, C., Jiang, Y., Zhang, Y., Wu, T., Jin, Q., Chanpaisit, N., et al. Vbench: Comprehensive benchmark suite for video generative models. In *Proceedings of the IEEE/CVF Conference on Computer Vision and Pattern Recognition*, pp. 21807–21818, 2024.

Kim, S. and Kim, H. A new metric of absolute percentage error for intermittent demand forecasts. *International Journal of Forecasting*, 32(3):669–679, 2016.

Kong, W., Tian, Q., Zhang, Z., Min, R., Dai, Z., Zhou, J., Xiong, J., Li, X., Wu, B., Zhang, J., et al. Hunyuanvideo: A systematic framework for large video generative models. *arXiv preprint arXiv:2412.03603*, 2024.

Li, C., Zhang, C., Xu, W., Lin, J., Xie, J., Feng, W., Peng, B., Chen, C., and Xing, W. Latentsync: Taming audioconditioned latent diffusion models for lip sync with syncnet supervision. *arXiv preprint arXiv:2412.09262*, 2024.

Li, R., Torr, P., Vedaldi, A., and Jakab, T. Vmem: Consistent interactive video scene generation with surfel-indexed view memory. *arXiv preprint arXiv:2506.18903*, 2025a.

Li, Z., Rahmani, H., Ke, Q., and Liu, J. Longdiff: Trainingfree long video generation in one go. In *Proceedings of the Computer Vision and Pattern Recognition Conference*, pp. 17789–17798, 2025b.

Lin, J., Yan, Y., and Wang, H. An end-to-end scene text detector with dynamic attention. In *Proceedings of the 4th ACM International Conference on Multimedia in Asia*, pp. 1–7, 2022.

Lin, J., Yan, Y., and Wang, H. A dual-path transformer network for scene text detection. In *ICASSP 2023-2023 IEEE International Conference on Acoustics, Speech and Signal Processing (ICASSP)*, pp. 1–5. IEEE, 2023.

Lin, J., Wu, Y., Wang, Z., Liu, X., and Guo, Y. Pair-id: A dual modal framework for identity preserving image generation. *IEEE Signal Processing Letters*, 2024a.

Lin, J., Zhao, G., Xu, J., Wang, G., Wang, Z., Dantcheva, A., Du, L., and Chen, C. Difftv: Identity-preserved thermalto-visible face translation via feature alignment and dualstage conditions. In *Proceedings of the 32nd ACM International Conference on Multimedia*, pp. 10930–10938, 2024b.

Lin, J., Zhang, C., Feng, W., Zhou, D., Wen, S., Du, L., and Chen, C. Apoavatar: Expressive audio-driven avatar generation via refocused audio-pose priors. 2025a.

Lin, S., Zhang, M., Chen, R., Wang, Q., Yang, X., and Geng, X. Linearly decomposing and recomposing vision transformers for diverse-scale models. *Advances in Neural Information Processing Systems*, 37:33188–33212, 2024c.

Lin, S., Yang, X., Wang, Q., Guo, S., Kou, Z., and Geng, X. Alpsb: Adaptive learngene with plastic and stable branches. *Pattern Recognition*, pp. 112623, 2025b.

Lin, S., Wang, Q., Liu, C., Yang, X., and Geng, X. Adaptivelearngene: Continual expansion and task-aware selection of learngenes for dynamic environments. In *Proceedings of the AAAI Conference on Artificial Intelligence*, volume 40, pp. 23576–23584, 2026a.

Lin, S., Wang, Q., Yang, X., and Geng, X. Unite: Universal knowledge integration from task-specific experts. In *International Conference on Learning Representations*, 2026b.

Lu, Y., Liang, Y., Zhu, L., and Yang, Y. Freelong: Trainingfree long video generation with spectralblend temporal attention. *Advances in Neural Information Processing Systems*, 37:131434–131455, 2024.

Ma, Y., Jin, T., Zheng, X., Wang, Y., Li, H., Wu, Y., Jiang, G., Zhang, W., and Ji, R. Ompq: Orthogonal mixed precision quantization. In *Proceedings of the AAAI Conference on Artificial Intelligence*, 2023a.

Ma, Y., Li, H., Zheng, X., Xiao, X., Wang, R., Wen, S., Pan, X., Chao, F., and Ji, R. Solving oscillation problem in post-training quantization through a theoretical perspective. In *Proceedings of the IEEE/CVF Conference on Computer Vision and Pattern Recognition*, 2023b.

Ma, Y., Li, H., Zheng, X., Ling, F., Xiao, X., Wang, R., Wen, S., Chao, F., and Ji, R. Affinequant: Affine transformation quantization for large language models. In *International Conference on Learning Representations*, 2024a.

Ma, Y., Li, H., Zheng, X., Ling, F., Xiao, X., Wang, R., Wen, S., Chao, F., and Ji, R. Outlier-aware slicing for post-training quantization in vision transformer. In *Proceedings of the 41st International Conference on Machine Learning*, 2024b.

Ma, Y., Zheng, X., Xu, J., Xu, X., Ling, F., Zheng, X., Kuang, H., Li, H., Wang, X., Xiao, X., Chao, F., and Ji, R. Flow caching for autoregressive video generation. In *International Conference on Learning Representations*, 2026.

Peebles, W. and Xie, S. Scalable diffusion models with transformers. In *Proceedings of the IEEE/CVF international conference on computer vision*, pp. 4195–4205, 2023.

Reinhard, E., Adhikhmin, M., Gooch, B., and Shirley, P. Color transfer between images. *IEEE Computer graphics and applications*, 21(5):34–41, 2002.

Song, Q., Wang, X., Zhou, D., Lin, J., Chen, C., Ma, Y., and Li, X. Hero: Hierarchical extrapolation and refresh for efficient world models. *arXiv preprint arXiv:2508.17588*, 2025a.

Song, Q., Zhou, D., Lin, J., Shen, F., Wang, J., Hu, X., Chen, C., and Heng, P.-A. Scenedecorator: Towards scene-oriented story generation with scene planning and scene consistency. *arXiv preprint arXiv:2510.22994*, 2025b.

Team, M. L., Cai, X., Huang, Q., Kang, Z., Li, H., Liang, S., Ma, L., Ren, S., Wei, X., Xie, R., et al. Longcat-video technical report. *arXiv preprint arXiv:2510.22200*, 2025.

Teng, H., Jia, H., Sun, L., Li, L., Li, M., Tang, M., Han, S., Zhang, T., Zhang, W., Luo, W., et al. Magi-1: Autoregressive video generation at scale. *arXiv preprint arXiv:2505.13211*, 2025.

Wang, C., Tian, K., Zhang, J., Guan, Y., Luo, F., Shen, F., Jiang, Z., Gu, Q., Han, X., and Yang, W. V-express: Conditional dropout for progressive training of portrait video generation. *arXiv preprint arXiv:2406.02511*, 2024a.

Wang, Z., Lin, J., Qian, Y., Huang, Y., Tian, S., Chai, B., Deng, J., Yang, Q., Du, L., Chen, C., et al. Diffx: Guide your layout to cross-modal generative modeling. *arXiv preprint arXiv:2407.15488*, 2024b.

Wu, B., Zou, C., Li, C., Huang, D., Yang, F., Tan, H., Peng, J., Wu, J., Xiong, J., Jiang, J., et al. Hunyuanvideo 1.5 technical report. *arXiv preprint arXiv:2511.18870*, 2025a.

Wu, H., Zhang, Z., Zhang, W., Chen, C., Li, C., Liao, L., Wang, A., Zhang, E., Sun, W., Yan, Q., Min, X., Zhai, G., and Lin, W. Q-align: Teaching lmms for visual scoring via discrete text-defined levels. *arXiv preprint arXiv:2312.17090*, 2023. Equal Contribution by Wu, Haoning and Zhang, Zicheng. Project Lead by Wu, Haoning. Corresponding Authors: Zhai, Guangtai and Lin, Weisi.

Wu, Y., Liang, S., Zhang, C., Wang, Y., Zhang, Y., Guo, H., Tang, R., and Liu, Y. From human memory to ai memory: A survey on memory mechanisms in the era of llms. *arXiv preprint arXiv:2504.15965*, 2025b.

Xiao, Z., Lan, Y., Zhou, Y., Ouyang, W., Yang, S., Zeng, Y., and Pan, X. Worldmem: Long-term consistent world simulation with memory. *arXiv preprint arXiv:2504.12369*, 2025.

Xie, D., Xu, Z., Hong, Y., Tan, H., Liu, D., Liu, F., Kaufman, A., and Zhou, Y. Progressive autoregressive video diffusion models. In *Proceedings of the Computer Vision and Pattern Recognition Conference*, pp. 6322–6332, 2025a.

Xie, Q., Ma, Y., Di, D., Gao, X., and Yang, X. Moca: Identity-preserving text-to-video generation via mixture of cross attention. In *Proceedings of the 7th ACM International Conference on Multimedia in Asia*, pp. 1–8, 2025b.

Xu, J., Zhu, A., Lin, J., Ke, Q., and Chen, C. Skeleton-ood: An end-to-end skeleton-based model for robust out-of-distribution human action detection. *Neurocomputing*, 619:129158, 2025.

Yang, S., Huang, W., Chu, R., Xiao, Y., Zhao, Y., Wang, X., Li, M., Xie, E., Chen, Y., Lu, Y., et al. Longlive: Real-time interactive long video generation. *arXiv preprint arXiv:2509.22622*, 2025a.

Yang, S., Kong, Z., Gao, F., Cheng, M., Liu, X., Zhang, Y., Kang, Z., Luo, W., Cai, X., He, R., et al. Infinitetalk: Audio-driven video generation for sparse-frame video dubbing. *arXiv preprint arXiv:2508.14033*, 2025b.

Yin, T., Zhang, Q., Zhang, R., Freeman, W. T., Durand, F., Shechtman, E., and Huang, X. From slow bidirectional to fast autoregressive video diffusion models. In *Proceedings of the Computer Vision and Pattern Recognition Conference*, pp. 22963–22974, 2025.

Yu, J., Bai, J., Qin, Y., Liu, Q., Wang, X., Wan, P., Zhang, D., and Liu, X. Context as memory: Scene-consistent interactive long video generation with memory retrieval. In *Proceedings of the SIGGRAPH Asia 2025 Conference Papers*, pp. 1–11, 2025a.

Yu, J., Qin, Y., Wang, X., Wan, P., Zhang, D., and Liu, X. Gamefactory: Creating new games with generative interactive videos. *arXiv preprint arXiv:2501.08325*, 2025b.

Zhang, K., Jiang, L., Wang, A., Fang, J. Z., Zhi, T., Yan, Q., Kang, H., Lu, X., and Pan, X. Storymem: Multi-shot long video storytelling with memory. *arXiv preprint arXiv:2512.19539*, 2025a.

Zhang, L., Cai, S., Li, M., Wetzstein, G., and Agrawala, M. Frame context packing and drift prevention in next-frame-prediction video diffusion models. *Advances in Neural Information Processing Systems*, 38:30546–30566, 2026.

Zhang, Z., Chang, S., He, Y., Han, Y., Tang, J., Wang, F., and Zhuang, B. Blockvid: Block diffusion for high-quality and consistent minute-long video generation. *arXiv preprint arXiv:2511.22973*, 2025b.

Zhang, Z., Dai, Q., Bo, X., Ma, C., Li, R., Chen, X., Zhu, J., Dong, Z., and Wen, J.-R. A survey on the memory mechanism of large language model-based agents. *ACM Transactions on Information Systems*, 43(6):1–47, 2025c.

Zheng, X., Ma, Y., Xi, T., Zhang, G., Ding, E., Li, Y., Chen, J., Tian, Y., and Ji, R. An information theory-inspired strategy for automated network pruning. *International Journal of Computer Vision*, 2025.

Zhou, D., Lin, J., Shen, G., Liu, Q., Gao, J., Liu, L., Du, L., Chen, C., Fu, C.-W., Hu, X., et al. Identitystory: Taming your identity-preserving generator for human-centric story generation. *arXiv preprint arXiv:2512.23519*, 2025a.

Zhou, D., Liu, G., Yang, H., Li, J., Lin, J., Huang, X., Liu, Y., Gao, X., Chen, C., Wen, S., et al. Omnishow: Unifying multimodal conditions for human-object interaction video generation. *arXiv preprint arXiv:2604.11804*, 2026.

Zhou, M., Ye, K., Shah, V., Mei, K., Delbracio, M., Milanfar, P., Patel, V. M., and Talebi, H. Reference-guided identity preserving face restoration. *arXiv preprint arXiv:2505.21905*, 2025b.

# A. Appendix

## A.1. LM-100K Dataset

To curate the LM-100K dataset, we processed a large-scale pool of private raw videos through a rigorous filtering and annotation pipeline. We first employed PySceneDetect (Contributors.) to ensure shot consistency and used Q-Align (Wu et al., 2023) to remove low-aesthetic material. To emphasize motion-intensive scenarios, we calculated optical flow between frames sampled at 4 fps, discarding videos with low average motion dynamics to ensure the final 100k clips contain significant physical activity. For annotation, each long-take video was partitioned into 5-second chunks, with keyframes from each segment processed by GPT-4o to generate fine-grained semantic captions. A second pass with GPT-4o was then used to align these chunk-level descriptions, ensuring a coherent and consistent storyline across the entire temporal horizon.

## A.2. Detailed Metrics

### A.2.1. PERCENTAGE-BASED ERRORS: MAPE AND WMAPE

Mean Absolute Percentage Error (MAPE) is a classic relative-error metric in forecasting and time-series evaluation (Kim & Kim, 2016; De Myttenaere et al., 2016), and similar percentage-normalized errors have also appeared in video quality related studies (Huang et al., 2020). Given predictions $\hat{y}_i$ and ground-truth values $y_i$ for $i = 1, \ldots, N$, MAPE is defined as

$$\text{MAPE} = \frac{100}{N} \sum_{i=1}^{N} \left| \frac{y_i - \hat{y}_i}{y_i} \right|. \tag{8}$$

Despite its interpretability, the division by $y_i$ makes MAPE unstable when $y_i$ is close to $0$.

A commonly used alternative is Weighted MAPE (WMAPE), which aggregates absolute errors and normalizes them by the total magnitude of the targets:

$$\text{WMAPE} = \frac{\sum_{i=1}^{N} |y_i - \hat{y}_i|}{\sum_{i=1}^{N} |y_i|}. \tag{9}$$

In our context, the same "relative deviation" principle can be applied to quantify how a video-level (or segment-level) quality score varies over time.

### A.2.2. VIDEO DRIFT ERROR (VDE)

Long video generation often exhibits a gradual accumulation of small artifacts: a minor quality shift within short temporal windows can compound into noticeable drift later in the sequence (Li et al., 2025b; Lu et al., 2024). To make such temporal degradation measurable, we introduce *Video Drift Error (VDE)*, a segment-wise drift penalty built on the normalization idea of WMAPE (Kim & Kim, 2016; De Myttenaere et al., 2016).

**Segmenting the timeline.** Let $V$ denote a video. We split $V$ into $N$ consecutive segments of equal duration,

$$V = \{S_1, S_2, \ldots, S_N\}, \tag{10}$$

where $S_1$ serves as the reference segment.

**Per-segment scoring.** Given a segment-level evaluator $g(\cdot)$ (e.g., a clarity score, a motion score, etc.), we obtain

$$m_i = g(S_i), \qquad i = 1, \ldots, N. \tag{11}$$

**Drift w.r.t. the reference segment.** We measure drift by the normalized deviation from the first segment:

$$r_i = \frac{|m_i - m_1|}{m_1}, \qquad i = 2, \ldots, N. \tag{12}$$

**Weighted accumulation along time.** To emphasize later segments (where drift is expected to accumulate), we optionally apply monotone weights $w_i$ (linear or logarithmic):

$$\text{VDE} = \sum_{i=2}^{N} w_i \, r_i, \qquad w_i \in \{\, N - i + 1, \, \log(N - i + 1) \,\}. \tag{13}$$

A larger VDE indicates stronger temporal instability of the measured attribute. Related "drift" penalties have also been explored for identity preservation and perceptual consistency, e.g., IP-FVR (Han et al., 2025) and MoCA (Xie et al., 2025b).

### A.2.3. METRIC-SPECIFIC VDE INSTANTIATIONS

With the VDE template above, each concrete metric specifies $g(\cdot)$ and thus $m_i$. For clarity, we write

$$\text{VDE}_{(\cdot)} = \sum_{i=2}^{N} w_i \frac{|m_i - m_1|}{m_1}, \qquad w_i \in \left\{ N - i + 1, \ \log(N - i + 1) \right\}. \tag{14}$$

**VDE Clarity ($\downarrow$).** This variant targets sharpness drift (e.g., progressive blur/defocus). Let $f_t \in S_i$ be frames in segment $S_i$ and $Y_t$ the luminance. We compute a per-frame sharpness proxy via Laplacian variance and average it within each segment:

$$m_i^{\text{clar}} = \frac{1}{|S_i|} \sum_{t \in S_i} \text{Var}(\nabla^2 Y_t), \qquad \text{VDE}_{\text{clar}} = \sum_{i=2}^{N} w_i \frac{|m_i^{\text{clar}} - m_1^{\text{clar}}|}{m_1^{\text{clar}}}. \tag{15}$$

**VDE Motion ($\downarrow$).** This variant captures temporal drift in motion magnitude or smoothness (e.g., late-stage jitter or freezing). Let $u_t$ be the optical flow between consecutive frames, and define motion energy as $E(u_t) = \|u_t\|_2$. A segment-level score can be formed by averaging energy over the segment:

$$m_i^{\text{mot}} = \frac{1}{|S_i| - 1} \sum_{t \in S_i} E(u_t), \tag{16}$$

or alternatively by averaging a motion-smoothness score $s_t$ (computed from inter-frame differences):

$$m_i^{\text{mot}} = \frac{1}{|S_i|} \sum_{t \in S_i} s_t. \tag{17}$$

The drift penalty is

$$\text{VDE}_{\text{mot}} = \sum_{i=2}^{N} w_i \frac{|m_i^{\text{mot}} - m_1^{\text{mot}}|}{m_1^{\text{mot}}}. \tag{18}$$

**VDE Aesthetic ($\downarrow$).** This measures whether global visual appeal (composition/color/lighting) remains stable over time. Given a learned frame-level aesthetic predictor $A(\cdot)$, we average within each segment:

$$m_i^{\text{aes}} = \frac{1}{|S_i|} \sum_{t \in S_i} A(f_t), \qquad \text{VDE}_{\text{aes}} = \sum_{i=2}^{N} w_i \frac{|m_i^{\text{aes}} - m_1^{\text{aes}}|}{m_1^{\text{aes}}}. \tag{19}$$

**VDE Background ($\downarrow$).** This variant focuses on background stability (e.g., flicker, texture boiling, unintended camera drift). Let $\mathbb{B}_t$ be the background mask and $u_t(x)$ the flow at pixel $x$. We define background staticness as the fraction of background pixels with motion below a threshold $\tau$:

$$\phi_t = \frac{1}{|\mathbb{B}_t|} \sum_{x \in \mathbb{B}_t} \mathbf{1}(\|u_t(x)\| \leq \tau), \qquad m_i^{\text{bg}} = \frac{1}{|S_i|} \sum_{t \in S_i} \phi_t, \tag{20}$$

and compute

$$\text{VDE}_{\text{bg}} = \sum_{i=2}^{N} w_i \frac{|m_i^{\text{bg}} - m_1^{\text{bg}}|}{m_1^{\text{bg}}}. \tag{21}$$

**VDE Subject ($\downarrow$).** This variant penalizes subject/identity drift (e.g., face morphing, attribute changes). Let $E(\cdot)$ be an identity encoder and $\bar{e}_1$ the mean embedding over subject crops in the reference segment $S_1$. We compute per-frame cosine similarity and then average within each segment:

$$s_t = \cos\big(E(\text{crop}_t), \bar{e}_1\big), \qquad m_i^{\text{subj}} = \frac{1}{|S_i|} \sum_{t \in S_i} s_t, \qquad \text{VDE}_{\text{subj}} = \sum_{i=2}^{N} w_i \frac{|m_i^{\text{subj}} - m_1^{\text{subj}}|}{m_1^{\text{subj}}}. \tag{22}$$

A.2.4. COMPLEMENTARY METRICS

In addition to drift-oriented measures, we report complementary long-video metrics following prior minute-scale evaluation protocols (Guo et al., 2025c; Cai et al., 2025). Concretely, we adopt five VBench dimensions (Huang et al., 2024) that are particularly relevant for long-horizon generation:

- *Imaging Quality*: frame-level technical fidelity (noise, blur, exposure artifacts), reflecting low-level visual integrity.

- *Motion Smoothness*: temporal continuity and realism of motion, discouraging jittery or discontinuous dynamics.

- *Aesthetic Quality*: perceptual attractiveness of frames, including composition, lighting, and color harmony.

- *Background Consistency*: stability of the scene background across time, capturing whether the environment remains coherent.

- *Subject Consistency*: stability of the main subject's appearance/identity across frames, measuring temporal coherence of the protagonist.

## A.3. Extra Computational Cost

We analyze the additional inference-time computation introduced by Semantic-Adaptive Hierarchical Memory (SAHM). SAHM incurs *no extra GPU-resident memory* for long-term storage since the global reservoir $\mathcal{M}_{global}$ is maintained on CPU/disk. The overhead comes from two parts: (i) global reservoir maintenance, and (ii) prompt-guided semantic retrieval.

**Offline cost (tokenization).** We pre-tokenize the text prompts once offline. Therefore, tokenization introduces no per-clip overhead during inference.

**(i) Global reservoir maintenance.** For each generated frame, we compute its SigLIP embedding and apply the similarity/quality gate (cosine similarity, aesthetic threshold, and blur check). This adds one lightweight encoder forward per frame plus negligible scalar checks. In contrast, the diffusion denoiser typically runs $S$ iterative denoising steps per frame/clip. Let $C_{\mathrm{enc}}$ and $C_{\mathrm{den}}$ denote the cost of one SigLIP forward and one denoiser forward, respectively. The relative overhead from reservoir maintenance is bounded by

$$\frac{F \cdot C_{\mathrm{enc}}}{F \cdot S \cdot C_{\mathrm{den}}} = \frac{C_{\mathrm{enc}}}{S \cdot C_{\mathrm{den}}}, \tag{23}$$

which is small for practical $S$ (and in practice $C_{\mathrm{enc}} \ll C_{\mathrm{den}}$ due to the denoiser operating on high-dimensional spatiotemporal tokens).

**(ii) Prompt-guided semantic retrieval.** When the prompt changes, we compute one prompt embedding and retrieve the top-$k$ most similar frames from $\mathcal{M}_{global}$. With embedding dimension $d$ and $N$ stored items in $\mathcal{M}_{global}$, a brute-force retrieval is a single matrix-vector similarity computation with complexity

$$\mathcal{O}(Nd) \text{ for similarities} \quad + \quad \mathcal{O}(N \log k) \text{ for top-}k \text{ selection.} \tag{24}$$

This is efficient even under conservative upper bounds on $N$.

**Upper/lower bounds for a 1-minute video.** Consider a 1-minute video at 24 fps, i.e., $F = 24 \times 60 = 1440$ frames. The number of stored frames $N$ depends on the sparsification rate:

- **Upper bound (store every frame):** $N_{\max} = 1440$.

- **Lower bound (store one frame every 5 seconds):** $N_{\min} = 60/5 = 12$.

Thus, the per-update retrieval cost lies in

$$Nd \in [12d, \ 1440d]. \tag{25}$$

For example, with a typical embedding size $d = 768$, this corresponds to $9.2 \times 10^3$ to $1.1 \times 10^6$ multiply-adds per retrieval update, which is negligible compared to a single denoiser forward.

**Total retrieval cost over the whole video.** Let the clip length be $\ell$ frames, so the number of clips is $C = F/\ell$. In the pessimistic setting where the prompt changes *every* clip and we store *every* frame, the reservoir size grows roughly linearly across clips: $N_i \approx i\ell$ at clip index $i$. The total number of similarity dot-products over the full minute is

$$\sum_{i=1}^{C} N_i \approx \sum_{i=1}^{C} i\ell = \ell \cdot \frac{C(C+1)}{2} = \frac{F^2}{2\ell} + \frac{F}{2}. \tag{26}$$

With $F = 1440$, this yields approximately $\frac{1,036,800}{\ell}$ dot-products. For a common setting $\ell = 16$, we get $64,800$ dot-products, i.e., $\approx 5.0 \times 10^7$ multiply-adds when $d = 768$. In the sparse regime ($N \leq 12$), the total dot-products are at most $12C$ (e.g., only 1080 when $\ell = 16$).

**Conclusion.** Even under extremely conservative assumptions (prompt changes every clip and storing every frame), the retrieval component is a small vector similarity computation. Moreover, $\mathcal{M}_{global}$ is typically far sparser than the upper bound due to the similarity/quality gate. Therefore, SAHM introduces only a minor computational overhead compared to the denoising backbone, while enabling long-horizon consistency.

