# OpenReview forum: "DynaMem: Consistent Long Video Generation via Hierarchical Memory and Motion Priors"
_ICML.cc/2026/Conference — ICML 2026 regular_

### Official Review · Reviewer_CRPr · 2026-02-25

**Soundness:** 3
**Presentation:** 3
**Significance:** 2
**Originality:** 2
**Overall Recommendation:** 4
**Confidence:** 3

**Summary:**

The paper tackles autoregressive long-video generation, where videos are synthesized clip-by-clip and suffer from semantic drift, motion decay, and appearance instability as length grows. It proposes DYNAMEM, a framework that leaves the diffusion backbone unchanged but adds three components: (1) Semantic-Adaptive Hierarchical Memory (SAHM), which maintains a sparse global reservoir of past frames and retrieves a prompt-conditioned working memory for conditioning; (2) Dynamics-Prioritized Optimization (DPO), combining gradient-based data selection and motion-weighted supervision to emphasize dynamic regions; and (3) Reference-Anchored Perceptual Alignment (RAPA), an inference-time Lab-space color-moment alignment to stabilize appearance. The paper also formalizes a Video Drift Error (VDE) metric family for measuring long-horizon degradation and evaluates on LV-Bench and VBench, reporting improved drift, quality metrics, human preference, and ablations for each module.

**Compliance With Llm Reviewing Policy:**

Affirmed.

**Final Justification:**

The authors have addressed my main questions. After reading the other reviewers’ comments and concerns, I remain conservative regarding the acceptance of this paper.

**Key Questions For Authors:**

1. The paper relies heavily on LV-Bench and a few qualitative examples, but lacks a breakdown of performance across different prompt types, such as complex multi-character interactions or drastic scene changes. The human study covers only 20 prompts, which seems limited given the wide range of long narratives and evolving prompts. Could the authors provide more comprehensive evaluation across various types of prompts, particularly those involving rapid content changes or conflicting semantics, to assess DYNAMEM's robustness?
2. While the paper highlights successes, there is little discussion of potential failure modes or negative outcomes. For example, how might SAHM retrieval introduce misleading memory, such as irrelevant objects from earlier scenes, or how might RAPA overcorrect and disrupt stylistic changes? Can the authors provide specific examples or analyses showing when and how these modules might negatively affect generation quality, especially under evolving prompts that intentionally alter appearance?
3. Section 3.3 mentions that influence functions are used to select videos whose gradients are similar to those of a motion-rich anchor set, but there are no details on how these gradients are computed, how similarity is measured, or how often this selection is updated. Could the authors clarify these aspects of DPO, and provide more details on how the gradient-based selection works? Additionally, can they isolate the effect of gradient-based selection from the motion-weighted loss through ablation studies to clarify which part of DPO contributes to the performance improvements observed in Table 4?

**Limitations:**

yes

**Strengths And Weaknesses:**

## Strengths
1. Clear articulation of the long-horizon failure modes and design goals. Figure 2 nicely illustrates appearance instability, motion decay, and error accumulation in existing autoregressive pipelines, and Figure 3 provides a reasonably clear end-to-end schematic of DYNAMEM, tying the three proposed components to these failure modes.
2. Modular architecture that does not modify the backbone. The method uses HunyuanVideo as a frozen or lightly fine-tuned DiT backbone and adds SAHM, DPO, and RAPA around it. This makes the approach in principle transferable to other diffusion transformers, which is practically attractive.
3. Empirically strong across multiple axes. On LV-Bench and VBench, DYNAMEM improves not only motion-related metrics but also subject and background consistency, and even aesthetic and image quality. This suggests genuine robustness improvements, not just trading one aspect for another.

## Weaknesses
1. Limited diversity and scale of evaluation scenarios explicitly reported. The paper leans heavily on LV-Bench plus a few qualitative examples (Figures 1, 3, 5, 6). There is no breakdown across different prompt types (e.g., complex multi-character interactions, drastic scene changes) or durations beyond the 1-minute IQA analysis. The human study covers only 20 prompts, which is nontrivial but limited given the huge space of long narratives and evolving prompts. It is unclear how DYNAMEM handles prompts with very rapid content changes or conflicting semantics.
2. No detailed investigation of failure modes or negative consequences of modules. The paper highlights successes but does not show scenarios where SAHM retrieval brings in misleading memory (e.g., earlier scenes with now-irrelevant objects), or where RAPA overcorrects and destroys intended stylistic changes. Without such analysis, it is hard to gauge when DYNAMEM might harm generation quality, especially under evolving prompts that deliberately change appearance.
3. Under-specified and somewhat hand-wavy description of DPO’s gradient-based data selection. Section 3.3 states that influence functions are used to choose videos whose gradients are similar to those of a motion-rich anchor set, but there are no details on how these gradients are computed, how similarity is measured, or how often this selection is updated. Without such details, DPO becomes partially irreproducible. Moreover, there is no ablation isolating the effect of gradient-based selection alone vs motion-weighted loss, so it is unclear which part of DPO is actually responsible for the gains in Table 4. This weakens the methodological clarity and scientific contribution of DPO.

---

> ### Author Rebuttal · Authors · 2026-03-31
>
> We sincerely appreciate the time you dedicated to reviewing our paper and the insightful feedback you provided. In the following, we present our responses to your comments.
>
> > Q1: Evaluation diversity and scale.
>
> **(1) Prompt-type diversity.**
>
> We address the robustness concern through three pieces of evidence: (a) the expanded human study below (50 prompts including multi-character and scene-transition cases), (b) a 1-minute multi-character demo, and (c) a failure mode analysis on challenging scene transitions. These show DynaMem's gains hold on complex scenarios, while we identify where it struggles (abrupt large-gap transitions).
>
> **(2) Human study expansion.**
>
> We expanded the human study from 20 to 50 prompts, including complex multi-character scenes and prompt transitions:
>
> | Compared method | Overall win % | ID/Scene win % | Motion win % |
> |----------------|--------------|----------------|-------------|
> | vs. BlockVid | 69.3 | 56.8 | 64.1 |
> | vs. SkyReels-V2 | 87.1 | 74.3 | 80.6 |
> | vs. LongLive | 75.9 | 68.1 | 71.6 |
>
> With 50 diverse prompts (vs. 20 in the paper), overall preference increases across all baselines. Gains are largest against SkyReels-V2 (87.1%) and LongLive (75.9%, up from 55.1%), driven by motion quality. Against BlockVid, the ID/Scene margin narrows on multi-character prompts where both methods face identity challenges, but overall preference still rises (63.9% → 69.3%).
>
> **(3) 1-minute generation test.**
>
> We provide a 1-minute generation of a multi-character indoor scene where a man and a woman repeatedly enter and exit the frame. Identity remains stable across re-appearances and motion stays fluid:
>
> [Demo Video](https://ams626.github.io/icml26/Dyna/1min-longvideo.mp4)
>
> Table 5 in the paper reports IQA over 10s–60s: with RAPA, quality stays at 1.97–1.99; without, it degrades from 1.98 to 1.67.
>
> > Q2: Failure modes.
>
> We tested kitchen → forest → riverside. The kitchen-to-forest transition (large scene gap) causes blending artifacts: kitchen elements leak into the forest because the reservoir is dominated by kitchen frames and retrieval returns low-similarity matches. Forest-to-riverside is smoother since the reservoir now contains closer frames. Even in the failure case, identity stays consistent and motion stays fluid:
>
> [Failure case video](https://ams626.github.io/icml26/Dyna/scene_trans_video.mp4)
>
> Limitation: DynaMem cannot handle very abrupt scene transitions in one step. Breaking large scene changes into intermediate prompts (kitchen → hallway → outdoor → forest) lets the reservoir accumulate bridging frames. We will note this in the revised paper.
>
> > Q3: DPO details and ablation.
>
> **(1) Gradient computation.**
>
> Prior methods filter by optical flow magnitude, conflating camera shake with meaningful motion. DPO instead asks: "does training on this video push the model in the same gradient direction as known-good motion videos?" This captures whether a sample teaches coherent dynamics, not just large pixel displacements. The anchor set uses optical flow (a noisy but cheap proxy); gradient similarity then identifies videos that teach coherent dynamics, including those with moderate flow but strong learning signal.
>
> Concretely: for each candidate video, we compute the diffusion loss gradient w.r.t. the last two DiT blocks. The anchor set is the top-500 videos by optical flow magnitude. We rank candidates by cosine similarity to the anchor set's mean gradient and select the top-40%. This is done once before training (8 GPU-hours on 32 H100s).
>
> **(2) Ablation separating gradient selection from motion-weighted loss.**
>
> | Variant | VDE Motion ↓ | VDE Subject ↓ | VBench Motion ↑ | VBench Subject ↑ |
> |---------|-------------|---------------|-----------------|------------------|
> | L_diff only | 0.0537 | 0.4125 | 0.9887 | 0.8856 |
> | + gradient selection only | 0.0165 | 0.1281 | 0.9956 | 0.9436 |
> | + motion-weighted loss only | 0.0298 | 0.2145 | 0.9947 | 0.9271 |
> | + both (full DPO) | **0.0096** | **0.0753** | **0.9956** | **0.9677** |
>
> Gradient selection alone provides the larger individual gain (VDE Motion: 0.0537 → 0.0165, a 3.3x reduction), confirming that choosing *what to train on* matters more than reweighting the loss. Motion-weighted loss alone also helps (0.0537 → 0.0298) but is less effective in isolation. Combining both yields the best results across all metrics. VBench Motion Smoothness saturates near its ceiling (0.996), so the additional benefit of motion-weighted loss is better captured by the more sensitive VDE Motion metric (0.0165 → 0.0096).

---

> > ### Author Rebuttal · Reviewer_CRPr · 2026-04-03
> >
> > Thank you for the authors’ response. Most of my questions have been addressed.

---

> > > ### Author Response · Authors · 2026-04-07
> > >
> > > Dear Reviewer,
> > >
> > > Thank you for your acknowledgement. We would like to express our sincere gratitude to you, as your helpful feedback and valuable suggestions have been instrumental in elevating the overall quality of our work.
> > >
> > > We will incorporate the relevant revisions, analyses, and results into the main paper or appendix. We are truly grateful for your time and effort, and we wish you all the best in your future research endeavors.
> > >
> > > Best regards,
> > >
> > > Authors of Paper 2925

---

### Official Review · Reviewer_D3Nw · 2026-03-04

**Soundness:** 3
**Presentation:** 3
**Significance:** 2
**Originality:** 2
**Overall Recommendation:** 4
**Confidence:** 5

**Summary:**

This paper proposes DYNAMEM, an autoregressive framework designed to mitigate semantic drift, motion decay, and appearance instability in long-form video generation by integrating hierarchical memory retrieval, motion-prioritized optimization, and perceptual color alignment.

**Compliance With Llm Reviewing Policy:**

Affirmed.

**Final Justification:**

The authors partially resolved my doubts, but I still maintain that Anchored Aware Alignment (RAPA) is a post-processing technique operating in the CIELAB color space and cannot be considered a core contribution of the paper. I will raise the rating to 4.

**Key Questions For Authors:**

* Algorithmic Distinction of SAHM: The Semantic-Adaptive Hierarchical Memory (SAHM) appears to be a standard application of Retrieval-Augmented Generation (RAG) principles to video frames. Given that similar memory-retrieval designs are already established in the literature (e.g., LongLive, FlowAcr-R1), could you clarify the specific architectural or conceptual innovations in SAHM that distinguish it from these existing frameworks? If the authors can demonstrate a unique mechanism for redundancy reduction or retrieval efficiency that leads to superior long-horizon consistency, it would significantly improve my assessment of the paper's novelty.

* Generalization and Ablation of RAPA: Reference-Anchored Perceptual Alignment (RAPA) is a post-processing technique operating in CIELAB space. Have the authors tested applying RAPA to the outputs of baselines like SkyReels or BlockVid? If RAPA alone resolves the "appearance instability" in these models, it suggests that DYNAMEM’s primary contribution may be a post-generation "trick" rather than a core improvement in the generative process.

* Dataset Bias: DYNAMEM is fine-tuned on the internal LM-100K dataset. In the quantitative comparisons shown in Table 1 , were the open-source baselines—such as MAGI-1, SkyReels-V2, and BlockVid—also retrained on LM-100K to ensure a level playing field? If the baselines were not retrained, the performance delta likely reflects the quality of the training data  rather than the efficacy of the SAHM or DPO components.

* Visual Evidence for DPO: The paper claims that Dynamics-Prioritized Optimization (DPO) counteracts motion decay. However, the supplementary demo materials do not appear to include direct, side-by-side video comparisons with the baselines mentioned in the text.

**Limitations:**

The authors have provided a dedicated Impact Statement addressing both the positive accessibility benefits and the risks of misuse for harmful content generation.

**Strengths And Weaknesses:**

Strengths
* Practical Efficiency: The design of the Semantic-Adaptive Hierarchical Memory (SAHM) is industrially pragmatic. By offloading the global visual reservoir to CPU/disk and using lightweight SigLIP embeddings for retrieval, the system manages to maintain long-term context without the prohibitive GPU memory overhead usually associated with long-context transformers

* Systematic Evaluation Framework: The introduction of Video Drift Error (VDE) is a sensible addition to the existing evaluation landscape. It effectively quantifies how specific attributes—like clarity or subject identity—degrade over extended temporal horizons, offering a more nuanced view than static, frame-based metrics.

Weaknesses

* Novelty of SAHM: The Semantic-Adaptive Hierarchical Memory (SAHM) is effectively a standard Retrieval-Augmented Generation (RAG) pipeline adapted for video frames. This design choice—using a global visual reservoir to provide long-term context —is a well-established paradigm in recent long-video literature. Specifically, models such as LongLive and FlowAcr-R1. As presented, the implementation of SAHM lacks the architectural or conceptual novelty required to distinguish it from these existing memory-based frameworks.

* Significance of RAPA: The Reference-Anchored Perceptual Alignment (RAPA) is fundamentally an inference-time heuristic that aligns perceptual color statistics in the CIELAB space. While it appears to mitigate appearance instability , it functions more as a post-processing trick than a core generative advancement. If applying this standalone alignment module to other baselines—such as SkyReels —yields similar stability gains, it significantly dilutes the perceived importance of the DYNAMEM generative framework as a whole.

* Verification of DPO Claims: Although the paper introduces Dynamics-Prioritized Optimization (DPO) as a specialized solution for motion decay , the supplementary demo materials lack direct, side-by-side visual comparisons with key open-source baselines. Without qualitative evidence contrasting DYNAMEM's motion quality against competitors in the demos , it is impossible to subjectively validate whether the framework actually provides the "stronger temporal dynamics"  it claims in the text.

---

> ### Author Rebuttal · Authors · 2026-03-31
>
> We sincerely appreciate the time you dedicated to reviewing our paper and the insightful feedback you provided. In the following, we present our responses to your comments.
>
> > Q1: SAHM novelty vs. LongLive/FlowAct-R1.
>
> SAHM and LongLive are architecturally quite different; grouping them under "RAG" obscures the distinction.
>
> **Different memory paradigms.** LongLive is a frame-level causal AR model where "memory" is implicit: the KV cache carries historical information, and a frame sink (the first frame chunk permanently retained in the cache) serves as a fixed global anchor. SAHM is an explicit external memory for clip-level bidirectional DiT. Generated frames are selectively stored in a CPU/disk reservoir and dynamically retrieved at each generation step.
>
> **Static anchor vs. dynamic retrieval.** LongLive's frame sink is always the first frame. If the video's content evolves far from the opening scene, this anchor becomes less relevant. SAHM's reservoir tracks the full semantic trajectory via compound gating (cosine sim < τ, aesthetic threshold, blur check), producing a sparse set of semantically distinct keyframes. Retrieval is prompt-guided: when the prompt evolves ("she walks into a garden" → "she sits on a bench"), the working memory shifts accordingly. LongLive handles prompt switches through KV-recache (recomputing the cache), which requires detecting explicit switch boundaries. SAHM handles both abrupt and gradual prompt evolution without switch detection.
>
> **Distinction from FlowAct-R1.** FlowAct-R1 uses a fixed-size memory buffer (1 reference image + 3 long-term latents + 1 short-term latent) populated by FIFO, with no selection and no retrieval; all slots always attend. This is essentially a sliding window with a static reference anchor. SAHM differs in both dimensions: *what enters memory* is filtered by semantic novelty and quality (not just recency), and *what comes out* is selected by prompt relevance (not "use everything").
>
> **Empirical validation.** Table 3 includes a "w/ all frames" baseline (naive retrieval from the full uncurated history, the closest analog to a standard RAG pipeline). SAHM achieves 2.4x lower VDE Subject (0.0753 vs 0.1845), confirming that compound gating and prompt-guided retrieval provide concrete gains over brute-force retrieval.
>
> We will add LongLive and FlowAct-R1 to the related work discussion in the revised paper.
>
> > Q2: RAPA generalization.
>
> We ran RAPA on two open-source baselines:
>
> | Method | VDE Aesthetic ↓ (w/o RAPA) | VDE Aesthetic ↓ (w/ RAPA) | VDE Subject ↓ (w/o) | VDE Subject ↓ (w/) |
> |--------|---------------------------|--------------------------|---------------------|---------------------|
> | SkyReels-V2 | 1.2083 | 0.3327 | 0.1085 | 0.1064 |
> | BlockVid | 0.9618 | 0.2194 | 0.0844 | 0.0772 |
> | DynaMem | 0.5428 | 0.0524 | 0.0977 | 0.0753 |
>
> RAPA generalizes: it reduces VDE Aesthetic by 3.6–4.4x on baselines and 10.4x on DynaMem. Two observations:
>
> First, RAPA benefits DynaMem disproportionately (10.4x vs 3.6–4.4x), and even after applying RAPA to all methods, DynaMem's VDE Aesthetic (0.0524) stays 4–6x lower than the best RAPA-augmented baseline (0.2194). This indicates SAHM and DPO create a better foundation for RAPA to work with; the modules are synergistic, not independent.
>
> Second, RAPA addresses a previously uncharacterized artifact: progressive saturation/contrast drift in autoregressive generation (Figure 6, Table 5). That it generalizes to other methods makes this a reusable finding, not a weakness.
>
> That said, RAPA alone cannot address motion decay. DynaMem's VDE Motion (0.0096) remains the lowest across all methods in Table 1, with or without RAPA; that advantage comes from DPO.
>
> > Q3: Dataset bias.
>
> All baselines in Table 1 use official released weights. This is standard: each method trains on its own (mostly private) data, and papers compare released models.
>
> The ablation (Tables 3–5) is more informative: every variant uses the same backbone and LM-100K data. The "w/ L_diff only" baseline in Table 4 uses LM-100K with standard diffusion loss and still suffers severe motion decay (VDE Motion 0.0537 vs 0.0096). The gap is not explained by data quality alone; the training objective and memory mechanism matter.
>
> > Q4: Visual evidence for DPO.
>
> We provide side-by-side comparisons with baselines (Self-Forcing, SkyReels-V2, FramePack, BlockVid): [Comparison Video](https://ams626.github.io/icml26/Dyna/side_by_side_baselines.mp4)
>
> In the video, DynaMem maintains consistent motion throughout the sequence. Baselines progressively lose dynamics or freeze into near-static imagery, while our method sustains coherent motion over the full 60s.

---

> > ### Author Rebuttal · Reviewer_D3Nw · 2026-04-02
> >
> > The authors partially resolved my doubts, but I still maintain that Anchored Aware Alignment (RAPA) is a post-processing technique operating in the CIELAB color space and cannot be considered a core contribution of the paper. I will raise the rating to 4.

---

> > > ### Author Response · Authors · 2026-04-02
> > >
> > > We thank the reviewer for the constructive feedback.
> > >
> > > Regarding RAPA's post-processing nature, we agree with the characterization. That said, many post-hoc techniques in generative modeling are recognized as contributions. Take classifier-free guidance as an example; it is applied after the core denoising and is widely adopted. RAPA also provides a plug-and-play fix that generalizes across architectures.
> > >
> > > ---
> > >
> > > Once again, thank you sincerely for engaging with our rebuttal and for the thoughtful questions throughout this discussion. We are truly grateful for the time and effort you have invested in reviewing our submission, and we hope the evidence and clarifications above address your remaining concern. We wish you all the best in your own research.
> > >
> > > Best regards,
> > >
> > > Authors of Paper 2925

---

### Official Review · Reviewer_xS5s · 2026-03-09

**Soundness:** 2
**Presentation:** 3
**Significance:** 3
**Originality:** 2
**Overall Recommendation:** 3
**Confidence:** 5

**Summary:**

This paper focuses on several common issues in long video generation, including semantic drift, motion decay, and appearance instability, and proposes a targeted framework named DYNAMEM.
Specifically, DYNAMEM introduces coordinated techniques spanning memory mechanisms, motion-prioritized optimization, and perceptual appearance anchoring to mitigate these problems. Experimental results demonstrate the effectiveness of the proposed modules in improving long video generation quality.

**Compliance With Llm Reviewing Policy:**

Affirmed.

**Final Justification:**

Thanks to the authors for the detailed responses. Most of my concerns have been addressed. However, considering the limited novelty of the work, the insufficient distinction from existing methods, and the lack of real-time performance, I will maintain my original score.

**Key Questions For Authors:**

Please see the above weaknesses.

**Limitations:**

Yes.

**Strengths And Weaknesses:**

### Strengths

(1) The paper addresses important challenges in long video generation, namely semantic drift, motion decay, and appearance instability, which have recently become critical issues in this area. Tackling these problems is of high research value.

(2) The paper is generally well written and reader-friendly, and the use of illustrative figures helps clearly explain the proposed approach.

### Weaknesses
Q1: Insufficient implementation details for several important components

(1) The proposed method is built upon HunyuanVideo 1.5, a foundation video generation model primarily designed for short video generation. However, the paper does not sufficiently explain how this model is extended to support long video generation in the proposed framework.

(2) When training on the LM-100K dataset, it is unclear what video duration is used during training. For example, are the training samples single 5-second clips paired with a prompt, or are longer sequences used? Since the paper states that motion decay mainly emerges in long-horizon settings, it would be helpful to clarify whether longer training sequences are necessary to mitigate this issue.

Q2: Limited methodological novelty

(1) The problems of error accumulation and motion decay in long video generation have recently attracted significant attention, and several related works have specifically addressed these issues, such as Self-Forcing++, Deep Forcing, and Reward Forcing. However, the paper lacks discussion and comparison with these approaches. In particular, Reward Forcing, which improves motion consistency through reinforcement learning, appears conceptually similar to the proposed approach and therefore deserves explicit comparison.

(2) The motivation behind the memory mechanism is somewhat unclear. The method may store frames that are aesthetically pleasing but inconsistent with previous frames, which could potentially harm temporal consistency in later generations. Although a top-k retrieval mechanism is used, it relies mainly on text-prompt semantic similarity, which may not effectively distinguish high-quality but inconsistent frames.

Q3: Additional experimental clarifications are needed

(1) The base model used in the paper, HunyuanVideo 1.5, appears to be around 8.3B parameters. However, many recent long video generation methods (e.g., Self-Forcing and LongLive) are built on Wan2.1-1.3B. Direct comparison with such models may therefore raise concerns regarding fairness due to the significant difference in model scale.

(2) Given that many existing long video generation methods are based on Wan2.1-1.3B, it would be valuable to examine whether the proposed method can also be applied to this model family.

(3) Regarding computational efficiency, Section A.3 provides some theoretical discussion, but quantitative results and comparisons with existing methods are missing. Providing empirical efficiency metrics would strengthen the evaluation.

---

> ### Author Rebuttal · Authors · 2026-03-31
>
> We sincerely appreciate the time you dedicated to reviewing our paper and the insightful feedback you provided. In the following, we present our responses to your comments.
>
> > Q1: Implementation details.
>
> **(1) How HunyuanVideo is extended to long video generation.**
>
> We use an autoregressive clip-by-clip loop. HunyuanVideo generates a single 5-second clip (~120 frames at 24fps). At each AR step, the last 9 frames from the previous clip serve as short-term motion context, and SAHM-retrieved frames provide long-term semantic context. Both are injected via channel concatenation into the denoising network. The DiT backbone is unchanged; only a post-training stage on LM-100K adapts the model to this conditioning scheme.
>
> **(2) Training duration and sequence length.**
>
> Each training sample is a clip pair: the current 5-second segment plus 9 context frames from the preceding segment, drawn from the same long-take video. The model learns to generate a clip conditioned on its context. Motion decay mitigation comes from DPO's gradient-based data selection and motion-weighted loss at the clip level, not from longer training sequences.
>
> We will add these details to Section 4.1.
>
> > Q2: Comparison with recent methods.
>
> **(1) Self-Forcing++, Deep Forcing, Reward Forcing.**
>
> These three methods operate in the **streaming autoregressive** paradigm (causal DiT + KV cache, typically on Wan2.1-1.3B), managing error accumulation at the frame/KV-cache level. DynaMem uses a different paradigm, **clip-by-clip autoregressive** generation (bidirectional DiT, HunyuanVideo), targeting inter-clip semantic drift, motion decay, and appearance instability. The two paradigms face different failure modes and the methods are largely complementary.
>
> - **Self-Forcing++** extends self-rollout to 100s with extended-DMD for long-horizon train-test alignment.
> - **Reward Forcing** uses EMA-Sink and reward-weighted distillation (Re-DMD) to improve motion dynamics in streaming generation.
> - **Deep Forcing** is training-free, using enlarged attention sinks and KV cache pruning.
>
> We compare with Reward Forcing (the most relevant open-source method targeting motion quality; Self-Forcing++ is not open-sourced, Deep Forcing is training-free and orthogonal):
>
> | Method | VDE Subject ↓ | VDE Background ↓ | VDE Motion ↓ | VBench Motion Smoothness ↑ |
> |--------|---------------|-------------------|--------------|---------------------------|
> | Reward Forcing | 0.0803 | 0.2854 | 0.0109 | 0.9956 |
> | DynaMem | **0.0753** | **0.2732** | **0.0096** | **0.9956** |
>
> Reward Forcing is built on Wan2.1-1.3B while DynaMem uses HunyuanVideo (8.3B), so this comparison reflects end-to-end system performance rather than a controlled ablation. Self-Forcing++ is not open-sourced; Deep Forcing is training-free and orthogonal to our approach. We will include all three in the related work discussion.
>
> **(2) Memory retrieval using text similarity.**
>
> Two design choices mitigate the risk of retrieving visually inconsistent frames:
>
> - Frames enter the reservoir only through a compound gate (cosine similarity < τ against the last stored frame, minimum aesthetic score, blur check). This produces a sparse, quality-filtered semantic timeline, not a collection of arbitrary "pretty frames."
> - Retrieved frames are injected as soft conditioning via channel concatenation; the denoiser can selectively ignore them. Table 3 confirms SAHM consistently improves all metrics.
>
> Adding visual similarity to retrieval (e.g., DINO features) could further help; we note this as future work.
>
> > Q3: Experimental fairness and efficiency.
>
> **(1) Model scale fairness.**
>
> We compare each method using its official released weights, which is standard practice (LongLive, SkyReels-V2, BlockVid do the same). The ablation study (Tables 3–5) isolates our modules on the same backbone and data, providing a controlled methodological assessment.
>
> To directly address scalability, we ran DynaMem's modules on Wan2.1-1.3B:
>
> | Method | VDE Subject ↓ | VDE Motion ↓ | VBench Subject Consistency ↑ |
> |--------|---------------|--------------|------------------------------|
> | Self Forcing baseline | 0.3716 | 1.6108 | 0.8481 |
> | Wan2.1-1.3B + DynaMem | 0.0819 | 0.2908 | 0.9615 |
>
> DynaMem's modules transfer to Wan2.1-1.3B with large gains, confirming the improvements are not backbone-dependent.
>
> **(2) Computational efficiency.**
>
> Inference cost for a 1-minute video (1440 frames, 24fps, H100, 30 steps, 5 seconds per clip):
>
> | Component | Time per clip | Memory overhead |
> |-----------|--------------|----------------|
> | Baseline (HunyuanVideo AR) | 65 s | 62 GB |
> | + SAHM (SigLIP + retrieval) | 1.8 s | ~0 GPU (CPU/disk) |
> | + RAPA (Lab alignment) | +<0.1 s | negligible |
> | DPO | training only | no inference cost |
>
> SAHM's overhead is small: SigLIP encoding is lightweight relative to DiT denoising, and retrieval is a single matrix-vector product on CPU.

---

> > ### Author Rebuttal · Reviewer_xS5s · 2026-04-03
> >
> > Thank you for the authors’ further response. Some of my previous concerns have been addressed, but several questions remain unresolved.
> >
> > Q1: Regarding the authors’ approach for extending HunyuanVideo to long video generation, DynaMem appears to function primarily as an incremental module, which limits the overall novelty of the work.
> >
> > Q2: Beyond the performance comparison with Reward Forcing, since Reward Forcing also improves motion consistency via reinforcement learning and is conceptually similar to the proposed method, a discussion of the technical implementation details is necessary.
> >
> > Q3: The authors did not address the issue of fairness arising from model scale differences. HunyuanVideo 1.5 (8.3B parameters) is substantially larger than existing long video generation models built on Wan2.1-1.3B, which may affect the fairness of the evaluation. Regarding computational efficiency, it is not sufficient to analyze only the components of DynaMem; comparisons with existing models (e.g., inference speed) are also required.
> >
> > Furthermore, since Wan2.1-1.3B only supports text-to-video (T2V) settings and does not accept video as reference input, it is necessary to clearly explain how DynaMem was adapted to Wan2.1-1.3B.

---

> > > ### Author Response · Authors · 2026-04-07
> > >
> > > We thank the reviewer for the continued engagement and address each point below.
> > >
> > > > Q1: Novelty for extending HunyuanVideo to long video generation.
> > >
> > > We appreciate the reviewer raising this point and would like to offer additional context. Building on existing backbones is common practice in long-video generation: FramePack (NeurIPS 2025 Spotlight) also extends HunyuanVideo. We believe the key question is what new capabilities a framework brings to the backbone it builds upon. FramePack is a particularly informative comparison because it shares DynaMem's paradigm (clip-level bidirectional generation). Yet on LV-Bench, which **emphasizes scene transitions and large-amplitude motion** over long horizons, Table 1 in our paper shows that FramePack exhibits substantially worse degradation:
> > >
> > > [FramePack Comparison](https://ams626.github.io/icml26/FramePack_Dyna_Comparison.png)
> > >
> > > FramePack mitigates forgetting and drifting through context compression and anti-drifting sampling, but does not address the compounding error chain we identify, which in turn triggers appearance instability. DynaMem breaks this chain at three stages: training (DPO), inference (SAHM), and output (RAPA). Each module targets a specific link in the chain rather than being independently motivated. Tables 3–5 confirm this: the full system outperforms any subset. To our knowledge, no prior work addresses these failure modes as a system.
> > >
> > > FramePack's Spotlight acceptance already shows that extending HunyuanVideo with targeted techniques is a recognized contribution of the work; DynaMem goes further by diagnosing a specific failure mode and addressing it step by step, with clear empirical advantages over FramePack and all other baselines.
> > >
> > > > Q2: Technical comparison with Reward Forcing.
> > >
> > > The technical mechanisms are quite different:
> > >
> > > - **Training**: Reward Forcing uses Re-DMD, a reward-weighted distribution matching distillation requiring a frozen teacher model. DynaMem uses direct supervised fine-tuning, no distillation.
> > > - **Motion mechanism**: Reward Forcing relies on VideoAlign as an external reward model to reweight DMD gradients. DynaMem uses influence functions to select training data and optical-flow saliency maps to reweight loss spatially, no reward model is needed.
> > > - **Memory**: EMA-Sink compresses evicted KV pairs into fixed-size tokens via exponential moving average (lossy). SAHM selectively stores frames on CPU/disk and retrieves by prompt similarity (lossless, scalable).
> > > - **Paradigm**: Streaming frame-level (causal DiT, few-step) vs. clip-level bidirectional (multi-step).
> > >
> > > They are complementary approaches for different deployment scenarios. We will add a detailed side-by-side table in the revised paper.
> > >
> > > > Q3: Model scale fairness, inference speed.
> > >
> > > **(1) Model scale fairness.**
> > >
> > > Baselines such as Self-Forcing are streaming methods (causal DiT + KV-cache, few-step distillation), so they choose Wan2.1-1.3B for better real-time performance. And methods like DynaMem and FramePack are clip-level bidirectional methods using other backbone (HunyuanVideo). They are all capable of long video generation, but with different emphases. Our goal is not real-time streaming, but rather pushing the upper bound of long video generation quality. All methods are trained with various strategies to achieve their best performance. Therefore, in Table 1, we include both real-time streaming and clip-by-clip approaches, and our method achieves state-of-the-art results.
> > >
> > > Furthermore, we have demonstrated that our method can be transferred to Wan2.1-1.3B with highly competitive performance:
> > >
> > > [Wan-Backbone Comparison](https://ams626.github.io/icml26/Transfer_Wan_1_3B.png)
> > >
> > > This confirms improvements stem from the framework, not model scale.
> > >
> > > **(2) Inference speed.**
> > >
> > > We list inference time for generating 50 seconds of video (1200 frames at 24fps) below:
> > >
> > > [Inference Speed Comparison](https://ams626.github.io/icml26/speed_comparison.png)
> > >
> > > Within the clip-level paradigm, DynaMem is **~4.5× faster** than FramePack. The speed gap between streaming and clip-level reflects the paradigm choice, not inefficiency. Clip-level bidirectional generation trades speed for capabilities such as: prompt-guided subject motion, temporal coherence, and lossless long-term memory (See [Long Demo Video](https://ams626.github.io/icml26/Dyna/1min-longvideo.mp4) in supplementary material).
> > >
> > > > Q4: Wan2.1-1.3B adaptation.
> > >
> > > We extend Wan2.1's first-layer input channels via channel concatenation to accept conditioning frames (last 9 frames + SAHM memory), with zero-initialization so pre-trained weights remain functional. We fine-tune on LM-100K with the same DPO strategy confirming cross-backbone transfer.
> > >
> > > ---
> > >
> > > Once again, thank you sincerely for engaging with our rebuttal and for the thoughtful questions throughout this discussion. We are truly grateful for the time and effort you have invested, and we hope the evidence and clarifications above can address your remaining concerns.

---

### Official Review · Reviewer_Quo3 · 2026-03-13

**Soundness:** 3
**Presentation:** 3
**Significance:** 2
**Originality:** 2
**Overall Recommendation:** 4
**Confidence:** 4

**Summary:**

This paper studies autoregressive long-video generation, where a model extends a video clip by clip and coherence degrades over long horizons. The authors proposed three components: (1) Semantic-Adaptive Hierarchical Memory, which stores a sparse long-term reservoir of past frames and retrieves prompt-relevant working memory; (2) Dynamics-Prioritized Optimization, which emphasizes motion-rich training samples and dynamic regions; and (3) Reference-Anchored Perceptual Alignment, which stabilizes appearance by aligning frame color statistics in CIELAB space. The system is implemented on top of a HunyuanVideo backbone, post-trained on an internal LM100K dataset of long-take videos, and evaluated using LVBench/VBench, human preference studies, and ablations. The paper reports improved subject/background consistency, motion stability, and appearance stability relative to several recent long-video baselines.

**Compliance With Llm Reviewing Policy:**

Affirmed.

**Final Justification:**

The rebuttal addressed my concerns and the additional qualitative comparisons are helpful, so I raised my score to 4.

**Key Questions For Authors:**

See weaknesses.

**Limitations:**

yes

**Strengths And Weaknesses:**

Strengths:
- The problem is important: long-horizon autoregressive video generation still suffers from semantic drift, motion decay, and appearance instability.
- The three compenents make sense: SAHM targets long-range semantics, DPO targets motion decay, and RAPA targets appearance drift.
- The paper is generally clear and easy to follow, especially in the high-level motivation and pipeline presentation.

Weaknesses:
- I strongly suggest the authors to rebrand the "DPO" part, because the name DPO is generally taken by Direct Preference Optimization in our community.
- The contribution is more a thoughtful integration of memory retrieval, motion-aware training, and appearance anchoring than a fundamentally new video generation framework. I personally think the paper is lack of a "core selling point".
- Are there only three video examples in the supp mat?

---

> ### Author Rebuttal · Authors · 2026-03-31
>
> We sincerely appreciate the time you dedicated to reviewing our paper and the insightful feedback you provided. In the following, we present our responses to your comments.
>
> > Q1: Rename "DPO".
>
> Agreed. We will rename it to Motion-Prioritized Optimization (MPO) throughout the paper.
>
> > Q2: Core selling point.
>
> We want to clarify the contribution more precisely.
>
> Semantic drift, motion decay, and appearance instability in autoregressive long-video generation are not three independent problems; they form a compounding error chain. Semantic drift causes the model to generate increasingly off-topic content; since the model has never been trained on such out-of-distribution states, dynamics collapse into static imagery (motion decay); static outputs then cause systematic pixel-distribution shift that accumulates frame over frame (appearance instability). Methods that target only one link show limited improvement because the other links keep feeding errors back.
>
> DynaMem breaks this chain at three different points in the pipeline:
>
> 1. **Training stage (MPO)** corrects the root data bias. Standard diffusion training is dominated by static backgrounds, so the model never properly learns temporal dynamics. MPO uses influence functions to select training videos whose gradient direction aligns with a motion-rich anchor set, distinguishing meaningful object motion from camera shake (unlike simple optical-flow filtering). Combined with spatially-adaptive loss weighted by motion saliency, this teaches the model to generate coherent motion rather than copying static context.
>
> 2. **Inference stage (SAHM)** provides long-range semantic anchoring that the limited context window cannot. The key design is not "retrieve past frames" (that is RAG), but *what enters memory* (compound gating produces a sparse, quality-filtered semantic timeline, not a raw history dump) and *what gets retrieved* (prompt-guided selection that adapts when prompts evolve).
>
> 3. **Output stage (RAPA)** corrects a systematic artifact we identified: autoregressive generation progressively shifts saturation and contrast, compounding over steps. RAPA applies CIELAB-space moment alignment anchored to early reference frames. The contribution is in identifying this specific perceptual drift pattern and providing a training-free, closed-form fix, analogous to how BatchNorm's contribution was identifying internal covariate shift, not the normalization operation itself.
>
> This three-stage decomposition also suggests a general recipe: autoregressive video systems can improve long-horizon coherence by intervening at data, inference, and output stages independently.
>
> > Q3: Supplementary videos.
>
> The original supplementary contained only three videos due to ICML's 100 MB attachment size limit. We provide additional demos via anonymous links:
>
> - Side-by-side comparisons with baselines (Self-Forcing, SkyReels-V2, FramePack, BlockVid):
>
> [Comparison Video](https://ams626.github.io/icml26/Dyna/side_by_side_baselines.mp4)
>
> - Ablation w/ vs w/o RAPA:
>
> [Ablation Video](https://ams626.github.io/icml26/Dyna/rapa_ablation.mp4)

---

> > ### Author Rebuttal · Reviewer_Quo3 · 2026-04-03
> >
> > Thank the authors for their rebuttal efforts. I will raise my score to a 4.

---

> > > ### Author Response · Authors · 2026-04-07
> > >
> > > Dear Reviewer,
> > >
> > > We sincerely appreciate your constructive comments and insightful questions regarding the details of our methodology, which have been very helpful in improving the quality and clarity of our work. We will incorporate all these clarifications into the revised paper. Thanks again for your invaluable comments.
> > >
> > > Best,
> > >
> > > Authors of Paper 2925

---

### Decision · Program_Chairs · 2026-04-30

**Decision:**

Accept (regular)

**Comment:**

This paper introduced three intuitive components to address three issues in long video generation: 1. SAHM for semantic drift; 2.  DPO for motion decay and 3.RAPA for appearance instability. The proposed method was tested on LV-Bench and VBench and demonstrate good performance.
Reviewers acknowledge the paper targets for an important problem and the overall design of the method is practical efficiency. They believe the experiments demonstrate strong empirical results on LV-Bench and VBench and the paper is well written. Most concerns raised by the reviewers were well addressed by the authors during the rebuttal and the paper got three weak accept rating. One reviewer still has concerns about the novelty and fairness comparison with other methods, based on the final response from the authors, I believe these concerns are well addressed and I tend to accept this paper.